**Subject Category:**
Biology (whole organism)

behaviour/ecology

singing, swimming speed, amount of singing, stamina, male quality

**Author for correspondence:**
Christopher W. Clark
e-mail: cwc2@cornell.edu

# Fin whale singing decreases with increased swimming speed

Christopher W. Clark[1,2], George J. Gagnon[2] and Adam S. Frankel[2]

[1]Bioacoustics Research Program, Cornell Lab of Ornithology, Cornell University, 159 Sapsucker Woods Road, Ithaca, NY 14850, USA
[2]Marine Acoustics, Inc., 2417 Camino Real South, Virginia Beach, VA 23456, USA

CWC, 0000-0002-7692-8150; ASF, 0000-0001-7154-1793

The attributes of male acoustic advertisement displays are often related to a performer's age, breeding condition and motivation, but these relationships are particularly difficult to study in free-ranging marine mammals. For fin whale singers, we examined the relationships between a singer's swimming speed, song duration and amount of singing. We used a unique set of fin whale singing and swimming data collected in support of the US Navy's marine mammal monitoring programme associated with the Navy's Integrated Undersea Surveillance System. A goal of the programme is to improve understanding of the potential effects of anthropogenic sound sources on baleen whale behaviours and populations. We found that as whales swam faster, some continued to sing, while others did not. If swimming speed is an indication of male stamina, then singing while swimming faster could be a display by which females and/or other males assess a singer's physical fitness and potential reproductive quality. Results have implications for interpreting fin whale singing behaviour and the possible influences of anthropogenic sounds on fin whale mating strategies and breeding success.

## 1. Introduction

Acoustic displays are subject to strong selection pressure when the level of display performance is related to signaller condition. In birds, anurans and mammals, conspicuous features of male acoustic displays can provide reliable cues of male attributes attractive to females and threatening to rival males, and there are known trade-offs between conspicuous acoustic features and other male traits [1–5]. In red deer, for example, minimum formant frequency is negatively correlated with an index of breeding success and thus an honest indication of male body size and fitness, and a salient feature by which females and

males assess males [6–8]. For free-ranging cetaceans, although there are no established methods by which to experimentally determine whether or not an acoustic performance attribute (e.g. amount of sound production, consistency, duration, frequency range) can be linked to reproductive success, we assume that insights into possible performance attributes can be deduced from empirical observations.

Fin whale (*Balaenoptera physalus*) singers have been identified as males, and the assumed biological function of song is either to attract females to mate, compete with male rivals for access to receptive females, or some combination of both [9,10]. A related question is whether or not these acoustic displays include some conspicuous, reliable feature(s) associated with male condition that a listening female and/or male might use to assess the 'quality' of the singer. For humpback whales (*Megaptera novaeangliae*), in which singers are also known to be males [11,12], several hypotheses have been suggested in which assessment of male singer quality involves measures of song characteristics (e.g. number of phrase types, song duration) assumed to be directly related to a singer's physical condition [13–15]. However, no hypothesized relationship between singing behaviour and male quality (e.g. physical stamina) for any baleen whale has been demonstrated with empirical data. In captive bottlenose dolphins, metabolic rates are higher during periods of vocal activity such that as vocal effort increases, there is a significant increase in metabolic rate, a result suggesting there are energetic consequences when wild animals attempt to compensate for increases in ambient noise levels due to anthropogenic noise [16].

Since the 1950s, the US Navy has maintained an extensive network of undersea hydrophone arrays known as the Sound Surveillance System (SOSUS) [17]. SOSUS was designed for monitoring the movements of submarines throughout vast areas of the Atlantic and Pacific oceans by detecting and tracking the sounds of individual vessels. SOSUS is now part of the larger Integrated Undersea Surveillance System (IUSS), which comprises fixed and mobile undersea acoustic sensors. Scientists have previously used SOSUS to detect, track and count the number of singing individuals for four endangered or protected whale species: blue (*Balaenoptera musculus*), fin, humpback and minke (*B. acutorostrata*) whales [18–20]. Each species sings species-specific songs either known or assumed to be male reproductive displays that occur seasonally throughout the North Atlantic and North Pacific oceans [9,21–24].

Fin whale song consists of repeated combinations of a few discrete, frequency-modulated and constant-frequency sounds, the most common of which is often referred to as the '20 Hz' note or pulse [22], even though the centre frequency of this note type is not necessarily 20 Hz (e.g. [25,26]). Applying the song terms used for humpback whales, in fin whales, a combination of regularly occurring units is a phrase, a sequence of repeated phrases is a song and a sequence of recorded songs is a song session [21].

Individual songs in a sequence are distinguished from each other by silent periods ('rests' [22]) that are often associated with the singer coming to the surface to breathe [9], and longer periods of silence between songs ('gaps' [22]) are assumed to occur when the animal is swimming from one locale to another or engaged in a different behaviour. Song is very intense, low-frequency, repetitive and stereotyped. Song source-level measurements have consistently been greater than 185 dB re 1 μPa @ 1 m (RMS) [27], and some amount of singing occurs throughout most of the year [10,20,22,28,29].

Using SOSUS, we observed and followed the singing behaviours of individual whales, from the very first song through the very last song in a singer's song sequence. We use the term 'singing bout' to refer to our observations of song sessions in which all songs in a singer's song sequence were observed and analysed. By this definition, all song data in this paper are based on singing bouts and do not include portions of a singer's song sequence.

Fin whale swimming speeds have been measured for singers and tagged whales. Singers in the North Pacific have been acoustically tracked while swimming between 1 and 14 km h$^{-1}$ [25,30]. In the North Atlantic, radio-tracked fin whales of unknown sex and unknown acoustic activity have been observed swimming between 10 and 16 km h$^{-1}$ for most of 4 days, and achieved speeds of 20 km h$^{-1}$ or greater for at least 20 min [31].

While using SOSUS assets to acoustically monitor and track fin whale singers in the North Atlantic, we noted that on many occasions, a singer transitioned between a period of singing relatively long, highly stereotyped songs and a period of either no singing or singing songs that appeared more variable in intensity and contained multiple, intermittent breaks in the song. We also observed that these different amounts and forms of singing were noticeably associated with a singer's swimming speed: when singers were swimming relatively slowly, they could sing highly stereotyped songs continuously except for regular 3–10 min breaks. When they started to swim faster, there was increased variability in a singer's singing behaviour: some stopped singing but started singing again

after they had moved to a new location, while some continued singing, but sang more variable songs as they swam more rapidly to a new location where they slowed down and resumed singing less variable songs. These observations piqued our interest and suggested that there was some reliable relationship between a male's swimming speed and his singing behaviour. Why would some males continue singing when they swam faster, and others would not? Was there something about this behaviour that was somehow adaptive and reliably indicative of a singer's physical stamina; something by which females could choose a higher quality mate and/or something by which competing males could decide whether or not to compete with or retreat from a singer, or instead were we observing a behaviour that served little to no benefit to the singer? Regardless of the eventual explanation, we knew that the ability of a singer to sing, even if only intermittent song, while swimming provided the only direct mechanism by which we could track a singer's speed, and therefore, we assumed that this was also the likely mechanism by which listening whales could assess a singer's speed. Given that energetic cost in marine mammals has been shown to be proportional to swimming speed to the third power [32], swimming fast is a physically demanding behaviour, which suggests that singing while swimming fast is a means by which conspecifics can possibly assess a singer's stamina.

Here, we show direct inverse relationships between individual fin whale swimming speeds and song durations and the proportion of time singing. These observations are consistent with the hypothesis, but not enough to conclude, that the ability of a singer to swim at higher speeds while continuing to sing has the potential to be an honest signal indicative of the singer's physical stamina and possibly his reproductive quality. Given reports demonstrating that some baleen whales modify their vocal behavioural rates and sound intensities as a result of anthropogenic noise [26,33–35], our results could have implications about the potential effects of anthropogenic sound sources on female choice and reproductive success in baleen whales.

# 2. Material and methods

Throughout April 2003 and April 2009, during 20 data collection exercises, we used Navy sensors installed during the Cold War to record, locate and track singing fin whales [19,20,22,24] throughout the North Atlantic Ocean during all months of the year. The system we used includes seafloor arrays containing multiple hydrophones that provide directional capabilities and acoustic gain, which enabled us to simultaneously and unambiguously detect, record, locate and track multiple individual singers for many days at a time throughout large areas of the ocean. IUSS sensors are still used today to track submarines and other ocean sound sources. Because the sensors are operational, details on specifications such as sensor location, water depth, and singer positioning and tracking resolutions cannot be provided (i.e. how different the features of tracks and acoustic signatures of two singers had to be for us to confidently maintain separation of their tracks).

We visually inspected singer tracks from multiple arrays to identify individual fin whale singers. We avoided periods when ambient noise conditions were elevated as a result of ocean noise and occasions when a seismic airgun survey was operating within or a surface vessel was transiting through the area in which a whale was singing. For each singing bout, we compiled the start and end times of each song in the bout and used time-of-arrival measurements for the same song on multiple arrays to calculate a series of singer locations throughout the singing bout. The path defined by a series of locations for an individual singer is referred to as a singer track or track. We used the time-varying features of each singer's track (i.e. speed and direction of travel), as well as the acoustic features observable in visual displays (e.g. spectrograms) containing each singer's songs, to unambiguously distinguish between different singers and tracks. Tracks also included information on the distance, bearing and speed between locations throughout an individual singer's track.

All tracks were mapped and carefully reviewed. If there was any ambiguity as to whether or not a track was derived from the same singer, that track was not included in further analysis. Thus, we have very high confidence that there was no ambiguity in the assignments of track locations to a single individual and that all locations in a track represent the movement of the same singer and are not mixtures of two or more singers. We are also very confident that the track data for each individual singer cover the entire period of that individual's singing bout(s). We assume that the same singer was not tracked more than once in the same year, but we cannot rule out this possibility, especially for the multi-year data. This process, which was designed to eliminate possibilities of confounding multiple tracks, does introduce some unknown level of bias against including tracks during periods of high singer density.

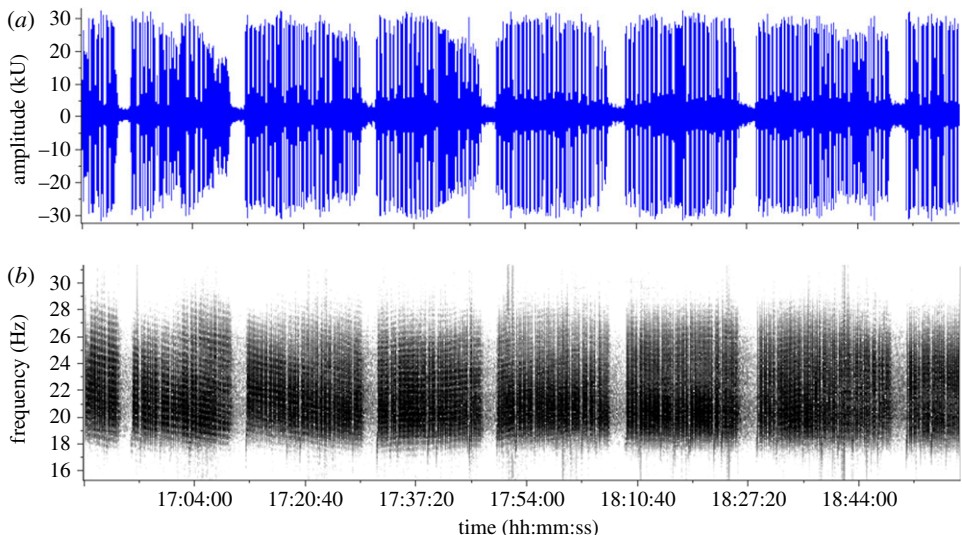

**Figure 1.** Robust songs from a fin whale swimming at less than 3 km h$^{-1}$ (11 January 2008) with an 88% duty-cycle. Amplitude, duration and inter-song silent gaps are considered regular and stereotypic. (*a*) Waveform (amplitude versus time). (*b*) A spectrogram (2 kHz sample rate, 16 384 pt. FFT, 50% overlap, Hann window). Note the duration is 2 h 12 min and see electronic supplementary material, Fig-S1_Fin-02_11Jan2008.wav. Note also that the black and white ribboning pattern in the figure is an example of acoustic interference in which the downward slope of the pattern indicates that the singer is moving away from the hydrophone and the upward pattern indicates that the singer is moving towards the hydrophone.

Based on previous studies, we presume that singers are males and that singing is a seasonal, reproductive display activity [9,22]. Our knowledge of the actual behavioural context for tracked singers is derived by inference and is limited, given that our acoustic data are not associated with any visual observations. Thus, we have no means by which to assess whether or not other whales responded to singers. We can only infer possible support for our assumptions by testing for relationships between variables within the data that are or are not consistent with the basic hypothesis that swimming faster while singing is a reliable proxy for male stamina and a possible indication of male quality.

Here, fin whale song is defined as a series of regularly spaced 20 Hz notes separated by regular periods of silence between successive notes. We define 'song duration' as the time period from the start of the first 20 Hz note in a song to the end of the last 20 Hz note in the song [22,26,28,29]. Our definition of song duration does not preclude the occurrence of other note types within a song (e.g. back-beats or 135–140 Hz downsweeps, [26]), nor does the occurrence of other note types affect the song duration metric. Lastly, we refer to the period of silence between successive singing bouts as inter-bout-intervals.

For the location and tracking data collection process, a minimum series of three, 20 Hz notes with regular inter-note-intervals of 9–20 s (typical of the present-day North Atlantic fin whale song) was required for location analysis and inclusion as a track position. By this criterion, two regularly spaced 20 Hz notes or single 20 Hz notes with unusually long, irregularly spaced inter-note-intervals (greater than 30 s) were not considered part of a song.

For each track, the distance travelled and the time interval between each pair of successive locations in the track were used to calculate a swim speed for each portion of a track (i.e. a track 'segment'). A duty-cycle metric, representing the amount of singing during a segment, was calculated as the total proportion of time the whale was singing during the last 30 min of the segment. Thus, a pair of swimming speed and associated duty-cycle values was calculated for each segment of a track. A duty-cycle greater than or equal to 70% was subjectively considered 'robust singing' (figure 1), and a duty-cycle less than 70% was subjectively considered 'intermittent singing' (figure 2) (see electronic supplementary material, acoustic data: Fig-S1_Fin-02_11Jan2008.wav and Fig-S2_Fin-01_17Feb2008.wav).

We define 'total-singing-duration' as the time span during which a whale was singing, from the start of the first song in the first bout to the end of the last song in the last bout. Thus, total-singing-duration is the same as bout duration for a whale that sang a single bout. We define 'track-duration' as the total amount of time between the first and last locations in a track and 'track-length' as the sum of all track segment distances in a track.

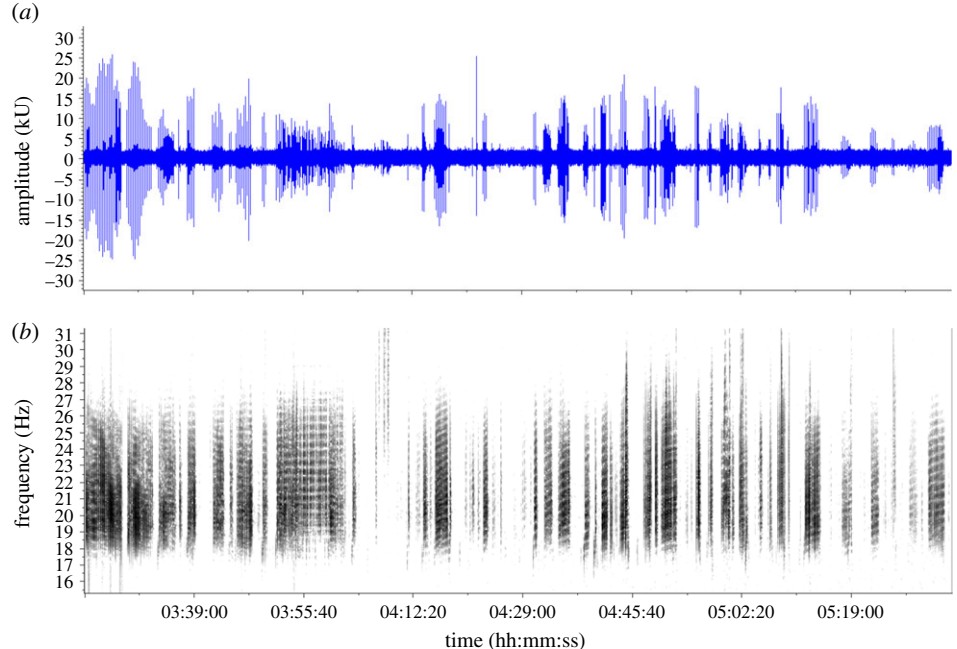

**Figure 2.** Less robust song from a fin whale swimming at more than 9 km h$^{-1}$ (17 February 2008) with a 48% duty-cycle. Amplitude, duration and inter-song silent gaps are considered highly variable. (*a*) The waveform (amplitude versus time). (b) The spectrogram (2 kHz sample rate, 16 384 pt. FFT, 50% overlap, Hann window). Note the duration is 2 h 22 min and see electronic supplementary material, Fig-S2_Fin-01-17Feb2008.wav.

Tracks are the unit of analysis, so that each whale is represented once in the analyses. For each track, the mean, standard deviation (s.d.) and minimum and maximum values of behavioural variables were calculated. Descriptive statistics for the entire set of tracks were generated from these mean track values. These statistics include the mean of means, standard error of the mean (s.e.m.) and means of minimum and maximum values.

For singers tracked for multiple bouts, tracks provide a means of comparing a singer's speeds while singing during a bout and not singing during the inter-bout-interval, but this comparison requires identifying a period of silence (i.e. inter-song-interval) between songs by which to decide when one bout ends and the next bout begins. To determine the bout ending criteria (BEC) value by which to define inter-bout-intervals, we used the methods described in Sibly *et al.* [36] and examined the log frequency distribution of all inter-song-interval values ($n = 7877$). A four-parameter, biexponential curve was fit to the log frequency distribution data in JMP 14.1 [37], and the curve fit parameter estimates were used to calculate the BEC [38], which was determined to be 35 min (see electronic supplementary material, Fig-S3.esm). By this criterion, a singer sang multiple bouts if there was an inter-song-interval in his track greater than or equal to 35 min. We tested for differences between an individual's average swimming speed while singing and while not singing using a matched-pairs *t*-test.

Relationships between the mean values of the dependent variables duty-cycle and song duration were examined as functions of predictor variables swimming speed, number of songs, day-of-year (DOY) and year using a generalized additive model (GAM), implemented in the R 'mgcv' package [39]. The distributions of mean song duration and mean swimming speed values were skewed; therefore, each distribution was transformed using the lognormal Johnson transform [40]. An identity link function and a Gaussian distribution were specified. Model fits were evaluated using the 'gam.check' function and visualized using the 'visreg' package [41]. The model was initially run with all predictor variables. Non-significant predictor variables were then removed one at a time until a statistically significant set of predictors for both dependent variables was achieved.

## 3. Results

A total of 163 singer tracks, including 1208 swimming speed and duty-cycle data pairs and 8040 song durations, were collected from August 2003 to April 2009. Each track had a minimum of three

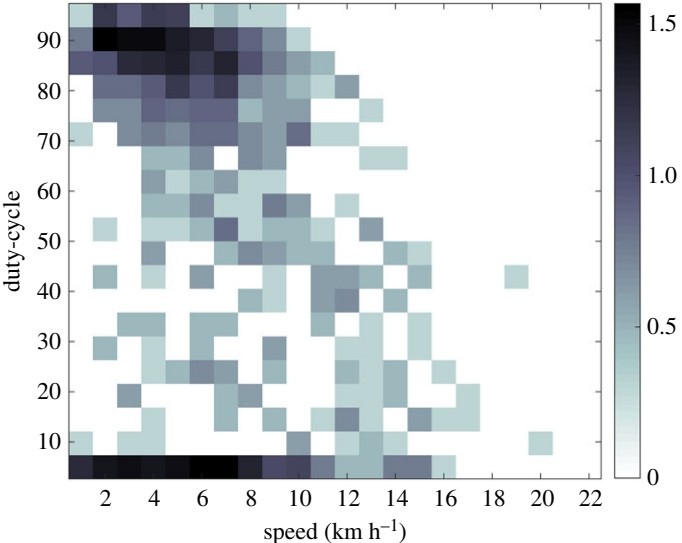

**Figure 3.** Three-dimensional histogram showing two modes of swimming behaviour based on 1208 track segment swimming speeds that comprise the 163 tracks. The major concentrations of swimming speeds for the low duty-cycle mode and high duty-cycle mode are primarily between 2 and 10 km h$^{-1}$. Note that cell count values have been converted to log$_{10}$ values, per the right-hand greyscale bar.

locations and a minimum duration of 1 h (see electronic supplementary material, table_S1). Successive locations in a track were at least 10 km apart, but we did not apply a minimum or maximum separation distance cut-off as the basis by which to decide if a location was part of the same track.

For all 163 tracks, the mean track-length was 86.2 km (s.d. = 89.8, median = 51, range = 10–430), mean track-duration was 14.9 h (s.d. = 14.7, median = 9.3, range = 1.1–70.2 h), mean total-singing-duration was 15.8 h (s.d. = 14.8, median = 10.3, range = 1.1–70.4 h), mean number of locations per track was 8.4 (s.d. = 7.1, median = 6, range = 3–39) and mean song duration was 9.8 min (s.d. = 5.6, median = 8.9, range = 0.7–25.3) (see electronic supplementary material, table_S2.esm).

The mean duty-cycle, including all non-singing periods, was 54.3% (s.d. = 21.9, median = 53.7, range = 9.2–95.0), and the mean track swimming speed was 6.7 km h$^{-1}$ (s.d. = 3.4, median = 6.2, range = 1.1–16.8). Based on this mean of track swimming speeds, and rounding that result to the nearest integer, we use 7 km h$^{-1}$ as the swimming speed at which we differentiate between swimmers that swam slower while singing (less than 7 km h$^{-1}$) and swimmers that swam faster while singing (greater than or equal to 7 km h$^{-1}$), while recognizing that this distinction is somewhat subjective and must not be used to overinterpret our results.

Most singers started their singing bouts at swimming speeds less than 7 km h$^{-1}$ (91, 57%) and either continued swimming at less than 7 km h$^{-1}$ (51, 55%), or switched back and forth between swimming faster and swimming slower during the track (33%), or switched to swimming faster and continued swimming faster for the remainder of their track (12%). For the singers that started their singing bouts swimming at greater than or equal to 7 km h$^{-1}$ (43%), most either continued swimming at greater than or equal to 7 km h$^{-1}$ (50%), or switched back and forth between swimming slower and swimming faster during the track (33%), or switched to swimming slower and continued swimming slower for the remainder of their track (17%). Although 57% of all singers started their tracks swimming at less than 7 km h$^{-1}$, only 31% of all singers continued swimming at less than 7 km h$^{-1}$ for their entire singing bout(s), while 69% of all singers swam at speeds greater than or equal to 7 km h$^{-1}$ during some period of their track.

Track segment speeds for all singers varied from less than 1 to 23 km h$^{-1}$ ($n = 1208$), and the mean segment speed for all singers was 6.4 km h$^{-1}$ (s.d. = 3.9, median = 5.7). The number of segments per track ranged from 2 to 38, while the average number of segments per track was 7.4 (s.d. = 7.1, median = 5). The full dataset of segment speeds was used to describe the frequency of occurrence of every combination of duty-cycle and swimming speed (figure 3). The distribution of track segment speeds shows a strong bimodal duty-cycle distribution: one is a high duty-cycle mode (75–90% duty-cycle range, 34% of track segments), with speeds distributed between 1 and 8 km h$^{-1}$. The second is the low duty-cycle mode (0–5% duty-cycle range, 21% of track segments), with speeds distributed primarily between 1 and 10 km h$^{-1}$.

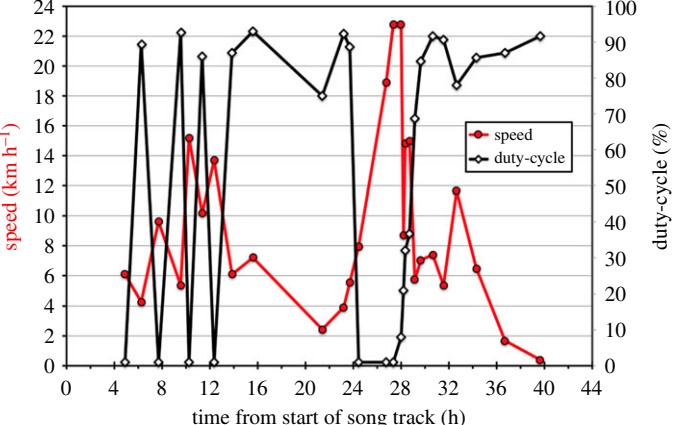

**Figure 4.** Song duty-cycle and swimming speed are shown as a function of time for Fin-05 on 13–15 January 2007 throughout his 39.6 h track starting at 21.14 GMT, 13 January and ending at 07.58 GMT, 15 January. Swimming speed (left y-axis) is shown as a red line. Duty-cycle (right y-axis) is shown as a solid black line. Note: 0.0 is the start of the singing bout, but the first measures of speed and duty-cycle in the track were obtained 4.9 h into the bout.

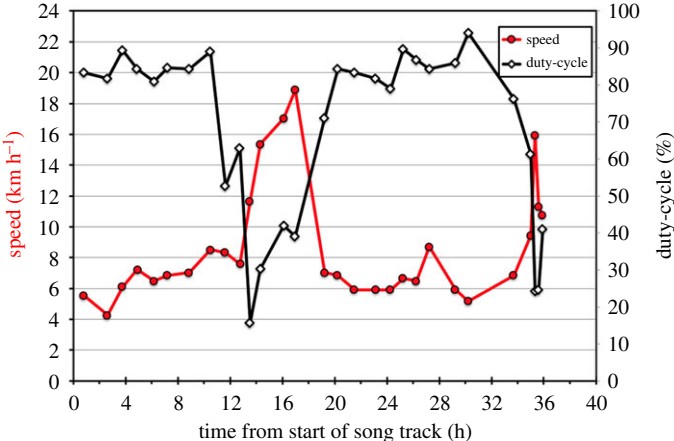

**Figure 5.** Song duty-cycle and swimming speed are shown as a function of time for Fin-10 throughout his 35.9 h track beginning at 15.36 GMT on 14 January 2007 and ending at 03.39 GMT on 16 January 2007. Swimming speed (left y-axis) is shown as a red line. Duty-cycle (right y-axis) is shown as a solid black line. Note: 0.0 is the start of the singing bout, but the first measures of speed (left y-axis) and duty-cycle (right y-axis) in the track were obtained 0.8 h into the bout.

Two examples illustrate the relationship between swimming speed and duty-cycle. Fin-05 on 13–15 January 2007 is notable due to the range of swimming speeds during his 39.6 h, 270 km track (figure 4). This is an example of a whale repeatedly alternating between singing with a high duty-cycle as a slower swimming singer (mean duty-cycle = 86.4%, s.d. = 6.90, mean swim speed = 5.7 km h$^{-1}$, s.d. = 2.88, $n = 16$), and switching to singing with a low duty-cycle as a faster swimming singer (mean duty-cycle = 9.5%, s.d. = 13.73, mean swim speed = 14.1 km h$^{-1}$, s.d. = 5.72, $n = 11$). This whale was able to maintain some level of singing at a maximum swimming speed of 15 km h$^{-1}$.

Fin-10 on 14–16 January 2007 (figure 5), a different individual to Fin-05 on 13–15 January because of their geographical separation, provides an example of a singer who swims slowly while singing with a high duty-cycle for long periods of time (mean duty-cycle = 76.1%, s.d. = 19.45, mean swim speed = 7.0 km h$^{-1}$, s.d. = 1.46, $n = 24$) and only once switches to fast swimming with a low duty-cycle (mean duty-cycle = 27.3%, s.d. = 14.10%; mean swim speed = 13.67 km h$^{-1}$, s.d. = 3.61, $n = 8$) over the course of his 35.9 h song bout and 285 km track. This whale was able to maintain some level of singing at a maximum speed of 19 km h$^{-1}$.

The examination of bout and inter-bout swimming speeds for 70 individual singers with an inter-song-interval greater than or equal to 35 min found that some sped up ($n = 33$) during the non-singing inter-bout period and some slowed down ($n = 37$) during the non-singing inter-bout period. Overall, the matched-pairs t-test found no significant difference in swimming speeds for individuals when singing and non-singing ($t_{69} = 0.002$, $p = 0.9985$).

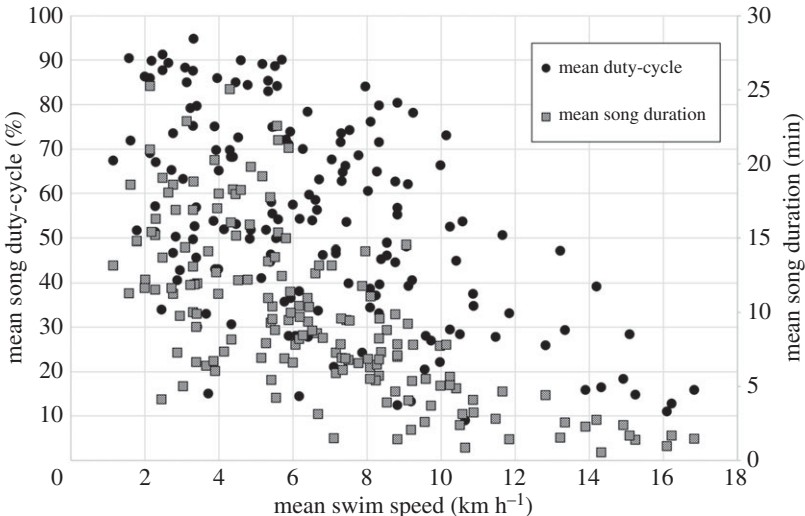

**Figure 6.** Mean duty-cycle (%) and mean song duration (min) as a function of mean swimming speed (km h$^{-1}$) for 163 fin whale singer tracks.

**Table 1.** The significance tests for the smoothing terms of each predictor variable for the duty-cycle model are presented. The degrees of freedom (d.f.) and the value of the *F*-ratio are shown, as well as the probability value. All three predictor variables were statistically significant.

| duty-cycle model | | | |
|---|---|---|---|
| variables | d.f. | *F* | *p*-value |
| whale speed | 3.1 | 21.94 | $<$0.001 |
| year | 4.7 | 3.67 | $<$0.01 |
| month | 4 | 21 | $<$0.001 |

The aggregate of all track data ($n = 163$) revealed obvious relationships between whale speed and both duty-cycle and song duration (figure 6). GAMs revealed that the overall model for duty-cycle explained significant amounts of variation (adjusted $r^2 = 0.57$) when the variables swimming speed, DOY and year were retained (table 1). Model curve fits for duty-cycle with these variables are shown in figure 7. The effects of swimming speed and DOY were quite strong, while yearly variation was less strong. The overall model for song duration retained the same set of variables as for duty-cycle (table 2), explained more variance (adjusted $r^2 = 0.69$) and the curve fits when the variables swimming speed, DOY and year were retained had similar shapes to those for duty-cycle (figure 8).

Figures 7 and 8 show that both duty-cycle and song duration decreased significantly as swimming speed increased. Both of these variables showed some degree of inter-annual variation, but it was not consistent for the two song parameters (figures 7*b* and 8*b*). However, our sampling dates varied between years, which may explain some or all of this inter-annual variability. Finally, there was a distinct seasonal peak during the September–February period for both duty-cycle and song duration (figures 7*b* and 8*b*).

## 4. Discussion

These results are based on an extensive set of acoustic observations derived from SOSUS assets that allowed us to record and track singers throughout entire singing bouts over a period of almost 6 years throughout the North Atlantic Ocean. These are the first results to report that as fin whale singers swim faster, they spend less time singing, and their song durations decrease. Several consistent characteristics of our results are worth emphasizing. All singers were swimming and changed swimming speeds throughout their singing bouts, and there was considerable variability in both swimming speed and the amount of singing within a bout. The relationship between swimming

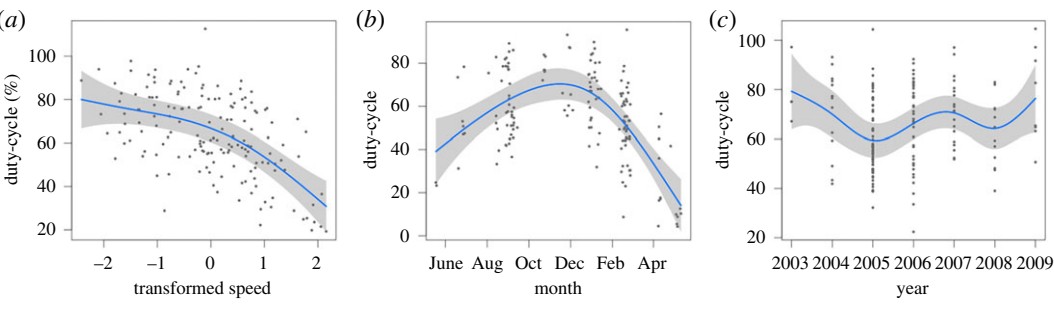

**Figure 7.** GAM model curve fits for duty-cycle for variables swimming speed (*a*), DOY (*b*) and year (*c*). The solid line shows the curve fit to the data and the grey shaded area around the curve shows the confidence limits. Data values are shown as dots. Duty-cycle decreases with swimming speed, has an obvious seasonal peak during the September to February period and shows some inter-annual variability.

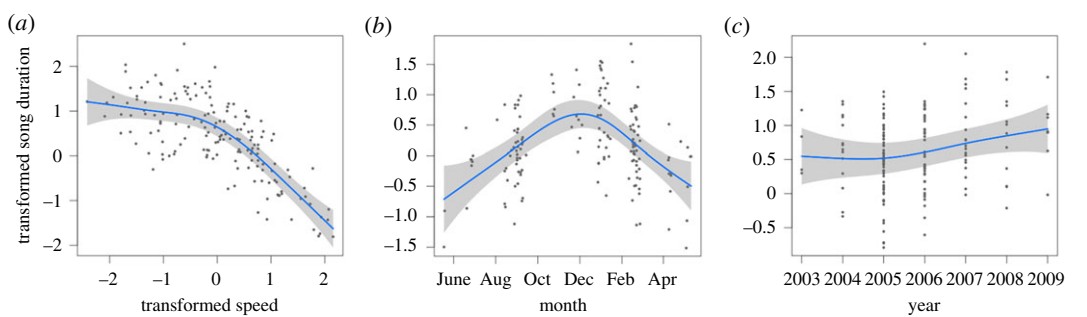

**Figure 8.** GAM model curve fits for song duration for variables swimming speed (*a*), DOY (*b*) and year (*c*). The solid line shows the curve fit to the data and the grey shaded area around the curve shows the confidence limits. Data values are shown as dots. Song duration decreases with swimming speed, has an obvious seasonal peak during the September to February period and shows some inter-annual variability that suggests that song duration is increasing with time.

**Table 2.** The significance tests for the smoothing terms of each predictor variable for the song duration model are presented. The degrees of freedom (d.f.) and the value of the *F*-ratio are shown, as well as the probability value. All three predictor variables were statistically significant.

| song duration model | | | |
|---|---|---|---|
| variables | d.f. | F | *p*-value |
| whale speed | 3.8 | 56.34 | <0.001 |
| year | 3.3 | 2.74 | <0.05 |
| month | 3.5 | 13.18 | <0.001 |

speed and song duty-cycle was bimodal, possibly representing different singer behavioural states as has been observed in humpbacks [42]. The ability to interpret the biological significance of these fin whale observations is especially challenging, given that 75% of our observations occurred from mid-September to mid-March throughout the pelagic North Atlantic, a time period and ocean area for which there are no associated or complementary visual observation reports.

By default, our interpretation of these results is placed within the context of a male reproductive display that serves either to attract females, compete with rival males, or some combination of both. These results are consistent with the general notion that baleen males use features of their singing behaviour as a mechanism by which to advertise their stamina and physical condition. For humpback whales, one hypothesis has been that song duration, which in some cases can be linked to breath-holding and directly related to a singer's physical condition, could be an indication of male quality [13]. Likewise, significant difference in song unit frequency and amplitude parameters between individuals has also been suggested as a potential indicator of singer quality [43]. For fin whales, a possible relationship between features of singing behaviour and physical stamina raise a number of

interesting questions as to the import of our empirical observations. Does the combination of singer attributes (swimming speed and amount of singing) represent an endurance behaviour with an unavoidable energetic cost, such that only males of high physical fitness can swim fast and continue singing? Does singing while swimming fast equate to an honest signal of male quality? If so, how does a female or male listener attend to these acoustically mediated attributes in order to assess singer quality? Are there singing-swimming strategies a singing male might adopt to increase a female or male listener's chances of assessing his physical fitness and potential reproductive quality? Are there strategies a female could adopt to increase her likelihood of selecting a high-quality mate or a male could adopt to counter a singer's display? Can our results provide any guidance as to the types of research that might address such questions? What is the possible import of these observations, given changes in marine acoustic environments as a result of anthropogenic sounds, including noises and signals?

Because our acoustic observation techniques provide no direct knowledge of the actual behavioural context or how listening whales might have responded to singers, we have no means by which to address these questions. However, in the future, one could specifically apply the acoustic technology used here to places and time periods when there are multiple singers within relatively close acoustic proximity to each other (i.e. less than 10 km). Under such conditions, one would be able to observe how singer–singer interactions are related to swimming and singing features. Clearly, the best way forward would be to apply multi-modal research approaches that include methods by which to observe both males and females (e.g. [44,45]).

Our results are consistent with but insufficient to assert our hypothesis that swimming speed is an indicator of a male's physical stamina (i.e. a performance display). We affirm that singing is the mechanism by which listeners can directly assess swimming speed, while assuming that listening whales can somehow judge differences in singer swimming speeds and that the combination of swimming while singing would provide the mechanism by which listeners assess a singer's physical fitness. Clark [46] emphasized that for extreme displays (e.g. male hummingbird flight display, male red deer roaring [47]), the energetic cost must include both the energy of short duration behavioural components and the power expended during the full duration of the behaviour. For fin whale singers, the energetic cost would be the quantity of energy per unit time expended during the behaviour of swimming, or swimming and singing combined, and the energetic cost would be metabolic power in watts or a correlate such as oxygen consumption rate [48].

For baleen whales, the modelled energetic cost of swimming in minke and humpback whales was found to be a function of swimming speed to the third power, and the authors imply that this relationship between speed and power is applicable to baleen whales in general [32,49]. This model thus predicts that relatively small increases in swimming speed impose relatively large increases in energetic/metabolic cost and that swimming speed is a direct and honest index of an energetic cost (i.e. faster swimming is a physically more costly behaviour).

The energetic cost of acoustic signalling has been well documented in amphibians and birds [50], and there is some evidence for vocalizing bottlenose dolphins (*Tursiops truncatus*) demonstrating a significant relationship between an increase in vocal effort and metabolic rate [16,51]. With regard to the energetic cost of singing, fin whale singers present an interesting situation. Songs consist of sequences of very intense 20 Hz notes [27,52]. A single song is typically composed from hundreds of high-energy notes and lasts for many minutes, singing bouts last for many hours, and singing occurs for more than six months of the year (figure 7b) [53,54]. Given the high level of singing effort in combination with the large muscle mass of the fin whale laryngeal system [55,56], it is not unreasonable to postulate that the energetic cost of singing in fin whales is relatively higher than other acoustic behaviours. Although singing is not necessarily a significant energetic factor, and we have no data by which to evaluate this possibility, the amount of singing while swimming, either in the form of duty-cycle or song duration, could also be a possible performance feature and an indication of a singer's quality.

Our suggestion that swimming speed might be a display feature advertising a singer's stamina has a number of challenging implications for listening conspecifics. To perceive and possibly assess singer swimming speed, a listening female or male must be able to hear singers and track their movements with enough resolution to judge variation in a singer's speed and discriminate between slower and faster swimmers. Although there is good evidence based on cochlear and auditory fibre measurements that fin whales have acute hearing in the very-low-frequency range (less than 100 Hz) [57,58], a fin whale's ability to track a singer's swimming speed through audition is entirely unknown, but must be acute if swimming speed provides an effective signal for assessing stamina. Thus, one is left to speculate how listeners assess swimming speed; for example, by attending to time-varying directional

cues (e.g. bearing rate) and time-varying changes in song characteristics (e.g. received level, frequency-dependent interference patterns, reverberation) as a result of sound transmission factors (e.g. water depth and variation in water temperature), but further speculation is beyond the scope of this paper.

During our extensive acoustic observations while tracking fin whale singers, only rarely have we observed what appeared to be singer–singer interactions based on the coordinated swimming and singing patterns of two animals. The maximum distances between singers during such observations have always been less than about 40 km. This does not mean that whales are not attending to or acoustically interacting with each other at greater distances, only that we have not been able to observe what might have appeared to us as acoustically mediated interactions at longer ranges. From a synthesis of our many years of tracking, we also estimate that about 80% of singers had fairly consistent directions of travel, while around 20% tended to meander within any particular area, a possible indication that singers were in different behavioural states [42]. We conservatively estimate that under typical ambient noise conditions in the 18–28 Hz band, a listener could hear and to some degree follow the movements of a singing male within a listening range of 10–50 km, while acknowledging there would be considerable variability in this range depending on ambient noise and sound transmission loss conditions in the 18–28 Hz frequency band, and the depths of the singing and listening whales.

A possible explanation for why some singers stopped singing as their swimming speeds increased is that they actively switched to another behaviour, as occurs in humpback whale males [59]. We cannot rule this out in fin whales, but this does not explain why the majority of singers (69%) sang at swimming speeds greater than $7\,\mathrm{km\,h^{-1}}$ for some portion of their song track, while less than one-third of singers (31%) only sang at swimming speeds less than $7\,\mathrm{km\,h^{-1}}$. If singers are successful at finding or attracting a female at higher swimming speeds, then why do nearly one-third of males only swim slower while singing? Are these younger or less physically fit males, or is this one of several strategies that males adopt depending on a context that we cannot yet observe? Given that swimming fast is energetically costly and possibly an indication of a male's physical stamina, this suggests that our results showing a highly significant inverse relationship between swimming speed and amount of singing is consistent with the conclusion, although highly speculative, that swimming speed can be an honest signal by which conspecifics assess male quality.

# 5. Conclusion

Our results, based on the application of US Navy SOSUS assets over a nearly 6-year period, demonstrate a significant relationship between a singer's swimming speed and amount of singing. Although the behavioural context for these acoustic observations are unknown, these results are consistent with the conclusion that the combination of swimming speed and amount of singing is a male reproductive display by which listening females and males can assess a singer's physical fitness and potential reproductive quality. If swimming speed and its association with singing is an honest signal of male quality, a man-made disturbance that causes a singer to swim faster and sing less has the potential to impose an artificial energetic cost on singers and reduce the amount of time engaged in the reproductive display, while also reducing the opportunities for listeners to assess singer quality. Based on our further observations using Navy assets to track different whale species, we strongly suspect that the relationship presented here between swimming speed and amount of singing for fin whales also occurs in other baleen whale singers. Whether for fin whales or other baleen species, this relationship between swimming speed and amount of singing has the potential to expose a possible mechanism by which to indirectly compare and assess the influences of natural factors as well as various anthropogenic activities on male reproductive activity and female choice.

Data accessibility. The swimming speed and associated duty-cycle data used in this manuscript are available in the electronic supplementary material as table_S1.esm. The acoustic wav files for figures 1 and 2 are available in the electronic supplementary material as Fig-S1_Fin-02_11Jan2008.wav and Fig-S2_Fin-01-17Feb2008.wav, respectively. The figure showing the curve fit used to calculate the inter-bout-interval of 35 min is available in the electronic supplementary material as figure_S1.esm.
Authors' contributions. G.J.G. made the initial observations, collected the primary data and completed the primary analysis. C.W.C. conducted the synthesis, writing and made some of the figures. A.S.F. conducted the statistical analyses, produced figures and assisted in the writing.
Competing interests. The authors have no competing interests.
Funding. Funding was provided by Chief of Naval Operations' Undersea Capabilities, N2/N6F24.

**Acknowledgements.** Thanks to the Chief of Naval Operations' Undersea Capabilities (N2/N6F24) Branch Head Captain Thomas Stanley; Deputy Branch Head Commander Cynthia Morgan; Commander Undersea Surveillance Captain Scott Rauch; and Commanding Officer, Naval Ocean Processing Facility, Dam Neck, VA Commander Jeffery Jacoby. Thanks also to Kathy Vigness-Raposa for providing valuable comments and review. Two anonymous reviewers provided stimulating and very helpful suggestions that improved the final manuscript.

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
