## [Reviewer comments · Royal Society Open Science]

Review History

RSOS-180525.R0 (Original submission)

Review form: Reviewer 1 (Eduardo Mercado III)

Is the manuscript scientifically sound in its present form?

Yes

Are the interpretations and conclusions justified by the results?

No

Is the language acceptable?

Yes

Is it clear how to access all supporting data?

No

Do you have any ethical concerns with this paper?

No

Have you any concerns about statistical analyses in this paper?

No

Recommendation?

Major revision is needed (please make suggestions in comments)

Comments to the Author(s)

See attached file (Appendix A).

Review form: Reviewer 2

Is the manuscript scientifically sound in its present form?

Yes

Are the interpretations and conclusions justified by the results?

No

Is the language acceptable?

Yes

Is it clear how to access all supporting data?

Yes

Do you have any ethical concerns with this paper?

No

Have you any concerns about statistical analyses in this paper?

No

Recommendation?

Accept with minor revision (please list in comments)

Comments to the Author(s)

General Comments:

This manuscript describes an apparent relationship between swimming speed and duty-cycle for singing fin whales. It is well written, with a variety of relevant background information provided. This work is interesting given the difficulty of studying breeding behavior in free-ranging fin whales, and inspires a range of additional questions worthy of future research.

The main issue with this manuscript is that the authors must be careful not to assert their speculations about honest signaling as demonstrated fact. While they have clearly shown that there is a correlation between swimming speed and singing, they have not demonstrated that this is an "endurance display" that is behaviorally meaningful to females or other males. The authors raise many interesting questions that may inspire new research, but it must be made clearer throughout that additional research is needed to confirm (or refute) their claims. While the work itself is sound, before this paper is suitable for publication the authors need to adjust the scope of the manuscript so as not to overextend the implications of their results.

Additional notes are provided in the Specific Comments section below, which should help to clarify and improve the manuscript.

Specific Comments:

P. 1, L. 15. Suggest "...improving understanding of the potential effects..."

P. 1, L. 16-17. Are the singers mentioned here known or presumed males?

P. 2, L. 37. Cetacean should read cetaceans.

P. 2, L. 34-38. This is true, but performance attributes really need to be linked to behavior or reproductive success to determine whether they are meaningful honest signals.

P. 3, L. 50-52. Are there any examples of this in better studied animals? That would strengthen the authors' argument.

P. 3, L. 54. Effects should be affects.

P. 3, L. 57-60. The authors are presenting an interesting hypothesis here, but it's one that needs further testing. There are several sentences in a row here with implications building on implications, and the authors should be careful not to overextend the results of their empirical finding.

P. 4, L. 72. Consist should be consists.

P. 4, L. 89-92. Yes, it is possible that the quality of song performance under higher swimming speeds contains information about male quality. However, whether or not this is true still needs to be addressed with further research. If this is found to be the case, the next relevant question becomes: is this meaningful to females?

P. 5, L. 96. Suggest "...higher speeds while continuing to sing has the potential to be an honest signal...". Just because something could be an honest signal does not mean that it is biologically relevant or meaningful to other individuals in the population (or that it is actually utilized as a cue by conspecifics).

P. 5, L. 108-109. It would be useful to provide some information about typical numbers of concurrent signers, how far apart they were, how long they were tracked on average, etc.

P. 5, L. 122. Suggest add comma after "through."

P. 6, L. 117. What is this "very high confidence" based on? The next sentence provides some clarification on this, but more details would be helpful.

P. 6, L. 121-122. How far apart did singers have to be in space and time to be considered separated from one another?

P. 6, L. 124-127. What do we know about patterns of swimming for this species generally? Is it likely to vary a lot over time (in which case an average across locations may obscure important details)? What about behavioral context?

P. 6, L. 136. Are the authors confident that all singing was captured? What proportion of vocalizations were thought to be recorded? How does ambient noise affect the range over which these calls are detectable by the hydrophone arrays?

P. 7, L. 142. What about variation in song duty-cycle and whale speed within a track? How was this accounted for?

P. 7, L. 153. How far apart were these locations typically? Was there a maximum distance allowed between two points to be considered the same track?

P. 7, L. 161-163. Are these values across the full tracks? Across two locations? Again, what was the variability like within a track?

P. 8, L. 167. What does "some singers" refer to? What proportion of singers exhibited this behavior? How long were they able to sustain this behavior?

P. 8, L. 170. Might song length be an alternative honest signal?

P. 8, L. 171. Why are the faster-swimming singers swimming faster? Are they really stopping singing because of energetic costs or because they are switching behaviors for some other reason? Context matters here, and these details are important to consider.

P. 8, L. 176. Suggest "...158 presumed unique singers..."

P. 8 - 9, L. 181-191. These examples are very useful. Can the authors comment on intra-individual variability in these patterns over time/season/context?

P. 9, L. 200-208. These are great questions. Additional data are needed to address them! The data presented herein do not allow the authors to answer any of these questions, although it is helpful

to discuss them and it would be great if the authors suggested some additional targeted research. P. 10, L. 241-243. This is not implied. Just because information is available does not mean that it can be reliably used by other individuals.

P. 10, L. 248. A listening individual is just one receiver. We don't know about the localization abilities of this species or how good a fin whale would be at evaluating the location and speed of another singer. The authors are making large assumptions about what is known and what is possible – these caveats must be acknowledged.

P. 10, L. 251-257. Do we have evidence that whales can actually do this?

P. 10, L. 244-259. There are lots of complicated, interacting factors presented here, and lots of assumptions. Again, the authors must be careful not to overextend their results.

P. 12, L. 274. What is this range based on? Is this under "typical" ambient noise conditions? How might auditory capabilities influence this?

P. 13, L. 284. Why are receptive females likely the most limiting resources for males? What is this based on? Do we know that female preference is the predominant driver of male behavior in fin whales?

P. 13, L. 294-297. This would be useful to mention or include a bit earlier (when listening distances are being discussed).

P. 13, L. 298-299. This may be true for this study, but this could be confirmed other ways (visual surveys, tagging, etc.).

P. 14, L. 312-313, 322-326. These are great points! The authors do a good job here of addressing some caveats of their assertions, but still need to be more explicit about this.

P. 14, L. 317-318. Maybe there are other reasons for this! Animals do not always perform optimally.

P. 14, L. 321. This is still speculative, and cannot be concluded directly from the observations/analysis in the present study.

P. 15, L. 347. Suggest "...ability of listeners to detect or assess the quality..."

P. 15, L. 351. Suggest "...duty-cycle might explain one way in which females and/or other males..."

P. 16, L. 355. This has not really been discussed in this way (i.e. that singing has a relatively lower energetic cost) previously in this paper, is this an important point?

P. 16, L. 364. It would be useful to suggest future research that could confirm or refute the hypothesis presented herein.

P. 21, Figures 1 and 2. Suggest noting duration of x-axis in the caption in each case so the reader can more easily see how to compare between the figures.

P. 21, Figures 3 and 4. Suggest adding (left y-axis) and (right y-axis) to the caption to denote swimming speed and duty-cycle.

P. 21, Figures 3 and 4. These examples are helpful!

Decision letter (RSOS-180525.R0)

24-May-2018

Dear Dr Clark,

The editors assigned to your paper ("Fin Whale Singing Decreases with Increased Swimming Speed") have now received comments from reviewers. We would like you to revise your paper in accordance with the referee and Associate Editor suggestions which can be found below (not including confidential reports to the Editor). Please note this decision does not guarantee eventual acceptance.

Please submit a copy of your revised paper within three weeks (i.e. by the 16-Jun-2018). If we do

not hear from you within this time then it will be assumed that the paper has been withdrawn. In exceptional circumstances, extensions may be possible if agreed with the Editorial Office in advance. We do not allow multiple rounds of revision so we urge you to make every effort to fully address all of the comments at this stage. If deemed necessary by the Editors, your manuscript will be sent back to one or more of the original reviewers for assessment. If the original reviewers are not available, we may invite new reviewers.

- Data accessibility

If you wish to submit your supporting data or code to Dryad (<http://datadryad.org/>), or modify your current submission to dryad, please use the following link:
<http://datadryad.org/submit?journalID=RSOS&manu=RSOS-180525>

- Competing interests

- Authors' contributions

- Acknowledgements

- Funding statement

Please note that Royal Society Open Science will introduce article processing charges for all new submissions received from 1 January 2018. Charges will also apply to papers transferred to Royal Society Open Science from other Royal Society Publishing journals, as well as papers submitted as part of our collaboration with the Royal Society of Chemistry (<http://rsos.royalsocietypublishing.org/chemistry>). If your manuscript is submitted and accepted for publication after 1 Jan 2018, you will be asked to pay the article processing charge, unless you request a waiver and this is approved by Royal Society Publishing. You can find out more about the charges at <http://rsos.royalsocietypublishing.org/page/charges>. Should you have any queries, please contact openscience@royalsociety.org.

Kind regards,
Andrew Dunn
Royal Society Open Science
openscience@royalsociety.org

on behalf of Dr Ari Friedlaender (Associate Editor) and Kevin Padian (Subject Editor)
openscience@royalsociety.org

Associate Editor's comments (Dr Ari Friedlaender):

To the Authors,

I have reviewed the comments from external reviewers and do not believe that in its current form the submission is acceptable for publication. The combination of too many unsubstantiated assumptions and a lack of support for major conclusions are concerning. Both reviewers agree that the conclusions presented are not justified by the results, and I concur. The reviewers provide very directed places where the authors need to reevaluate their work and make significant modifications, as one reviewer notes the major conclusions are 'based on a number of unwarranted assumptions rather than on any evidence'. The number of times the reviewers find unsupported conclusions, speculation or statements beyond what the results can suggest given the number of assumptions being made are significant.

The authors also fail to provide reasonable information on the methodology, assumptions and limitations of the manner in which the vocalizations were recorded. This needs to be adjusted in order to be acceptable for publication.

I encourage the authors to provide point by point responses to all the concerns expressed by the reviewers, explain their stance in each case, and incorporate these into the language of the submission.

Thank you.

Ari S. Friedlaender

Associate Editor: 2

Comments to the Author:

(There are no comments.)

Comments to Author:

Reviewers' Comments to Author:

Reviewer: 1

Comments to the Author(s)

See attached file

Reviewer: 2

Comments to the Author(s)

General Comments:

This manuscript describes an apparent relationship between swimming speed and duty-cycle for singing fin whales. It is well written, with a variety of relevant background information provided. This work is interesting given the difficulty of studying breeding behavior in free-ranging fin whales, and inspires a range of additional questions worthy of future research.

The main issue with this manuscript is that the authors must be careful not to assert their speculations about honest signaling as demonstrated fact. While they have clearly shown that there is a correlation between swimming speed and singing, they have not demonstrated that this is an "endurance display" that is behaviorally meaningful to females or other males. The authors raise many interesting questions that may inspire new research, but it must be made clearer throughout that additional research is needed to confirm (or refute) their claims. While the work itself is sound, before this paper is suitable for publication the authors need to adjust the scope of the manuscript so as not to overextend the implications of their results.

Additional notes are provided in the Specific Comments section below, which should help to clarify and improve the manuscript.

Specific Comments:

P. 1, L. 15. Suggest "...improving understanding of the potential effects..."

P. 1, L. 16-17. Are the singers mentioned here known or presumed males?

P. 2, L. 37. Cetacean should read cetaceans.

P. 2, L. 34-38. This is true, but performance attributes really need to be linked to behavior or reproductive success to determine whether they are meaningful honest signals.

P. 3, L. 50-52. Are there any examples of this in better studied animals? That would strengthen the authors' argument.

P. 3, L. 54. Effects should be affects.

P. 3, L. 57-60. The authors are presenting an interesting hypothesis here, but it's one that needs further testing. There are several sentences in a row here with implications building on implications, and the authors should be careful not to overextend the results of their empirical finding.

P. 4, L. 72. Consist should be consists.

P. 4, L. 89-92. Yes, it is possible that the quality of song performance under higher swimming speeds contains information about male quality. However, whether or not this is true still needs to be addressed with further research. If this is found to be the case, the next relevant question becomes: is this meaningful to females?

P. 5, L. 96. Suggest "...higher speeds while continuing to sing has the potential to be an honest signal...". Just because something could be an honest signal does not mean that it is biologically relevant or meaningful to other individuals in the population (or that it is actually utilized as a cue by conspecifics).

P. 5, L. 108-109. It would be useful to provide some information about typical numbers of concurrent signers, how far apart they were, how long they were tracked on average, etc.

P. 5, L. 122. Suggest add comma after "through."

P. 6, L. 117. What is this "very high confidence" based on? The next sentence provides some clarification on this, but more details would be helpful.

P. 6, L. 121-122. How far apart did singers have to be in space and time to be considered separated from one another?

P. 6, L. 124-127. What do we know about patterns of swimming for this species generally? Is it likely to vary a lot over time (in which case an average across locations may obscure important details)? What about behavioral context?

P. 6, L. 136. Are the authors confident that all singing was captured? What proportion of vocalizations were thought to be recorded? How does ambient noise affect the range over which these calls are detectable by the hydrophone arrays?

P. 7, L. 142. What about variation in song duty-cycle and whale speed within a track? How was this accounted for?

P. 7, L. 153. How far apart were these locations typically? Was there a maximum distance allowed between two points to be considered the same track?

P. 7, L. 161-163. Are these values across the full tracks? Across two locations? Again, what was the variability like within a track?

P. 8, L. 167. What does "some singers" refer to? What proportion of singers exhibited this behavior? How long were they able to sustain this behavior?

P. 8, L. 170. Might song length be an alternative honest signal?

P. 8, L. 171. Why are the faster-swimming singers swimming faster? Are they really stopping singing because of energetic costs or because they are switching behaviors for some other reason? Context matters here, and these details are important to consider.

P. 8, L. 176. Suggest "...158 presumed unique singers..."

P. 8 - 9, L. 181-191. These examples are very useful. Can the authors comment on intra-individual variability in these patterns over time/season/context?

P. 9, L. 200-208. These are great questions. Additional data are needed to address them! The data presented herein do not allow the authors to answer any of these questions, although it is helpful to discuss them and it would be great if the authors suggested some additional targeted research.

P. 10, L. 241-243. This is not implied. Just because information is available does not mean that it can be reliably used by other individuals.

P. 10, L. 248. A listening individual is just one receiver. We don't know about the localization abilities of this species or how good a fin whale would be at evaluating the location and speed of another singer. The authors are making large assumptions about what is known and what is possible - these caveats must be acknowledged.

- P. 10, L. 251-257. Do we have evidence that whales can actually do this?
- P. 10, L. 244-259. There are lots of complicated, interacting factors presented here, and lots of assumptions. Again, the authors must be careful not to overextend their results.
- P. 12, L. 274. What is this range based on? Is this under “typical” ambient noise conditions? How might auditory capabilities influence this?
- P. 13, L. 284. Why are receptive females likely the most limiting resources for males? What is this based on? Do we know that female preference is the predominant driver of male behavior in fin whales?
- P. 13, L. 294-297. This would be useful to mention or include a bit earlier (when listening distances are being discussed).
- P. 13, L. 298-299. This may be true for this study, but this could be confirmed other ways (visual surveys, tagging, etc.).
- P. 14, L. 312-313, 322-326. These are great points! The authors do a good job here of addressing some caveats of their assertions, but still need to be more explicit about this.
- P. 14, L. 317-318. Maybe there are other reasons for this! Animals do not always perform optimally.
- P. 14, L. 321. This is still speculative, and cannot be concluded directly from the observations/analysis in the present study.
- P. 15, L. 347. Suggest “...ability of listeners to detect or assess the quality...”
- P. 15, L. 351. Suggest “...duty-cycle might explain one way in which females and/or other males...”
- P. 16, L. 355. This has not really been discussed in this way (i.e. that singing has a relatively lower energetic cost) previously in this paper, is this an important point?
- P. 16, L. 364. It would be useful to suggest future research that could confirm or refute the hypothesis presented herein.
- P. 21, Figures 1 and 2. Suggest noting duration of x-axis in the caption in each case so the reader can more easily see how to compare between the figures.
- P. 21, Figures 3 and 4. Suggest adding (left y-axis) and (right y-axis) to the caption to denote swimming speed and duty-cycle.
- P. 21, Figures 3 and 4. These examples are helpful!

Author's Response to Decision Letter for (RSOS-180525.R0)

See Appendix B.

RSOS-180525.R1 (Revision)

Review form: Reviewer 1 (Eduardo Mercado III)

Is the manuscript scientifically sound in its present form?

Yes

Are the interpretations and conclusions justified by the results?

No

Is the language acceptable?

Yes

Is it clear how to access all supporting data?

Yes

Do you have any ethical concerns with this paper?

No

Have you any concerns about statistical analyses in this paper?

No

Recommendation?

Accept with minor revision (please list in comments)

Comments to the Author(s)

See attached file (Appendix C).

Review form: Reviewer 2

Is the manuscript scientifically sound in its present form?

Yes

Are the interpretations and conclusions justified by the results?

Yes

Is the language acceptable?

Yes

Is it clear how to access all supporting data?

Yes

Do you have any ethical concerns with this paper?

No

Have you any concerns about statistical analyses in this paper?

No

Recommendation?

Accept with minor revision (please list in comments)

Comments to the Author(s)

Re-Review of "Fin Whale Singing Decreases with Increased Swimming Speed"
Royal Society Open Science

In this revision, the authors have taken care to address the concerns of myself and the other original reviewer. As before, this work clearly demonstrates systematic differences in fin whale song production depending on swimming speed. The additions related to song duration strengthen the data set. The work is important and scientifically sound, and the analyses are appropriate.

The suggestion that listeners can assess male fitness based on swimming speed (with singing as the mechanism for determining speed) is interesting. While still speculative, the authors do a much better job of presenting their hypothesis as just that, and acknowledging their assumptions. Whether song production might also be an indicator of male quality remains to be seen, and the manuscript is simpler and more compelling without this additional assertion. I do think the discussion section could be shortened and made even simpler/more concise for clarity, with more emphasis on the direct meaning of the results rather than just the interesting potential implications.

Based on the authors' revisions, I can now recommend this manuscript for publication.

A few specific comments follow:

P 8, L 166. Effect should be affect.

P 8, L 178-179. Why was a duty cycle >70% defined as "robust?" Was this an arbitrary distinction?

P 9, L 198-200. Should this information be provided in the Results rather than the Methods section?

P 9, L 208. Remove comma after "initially."

P 11, L 251. As I am reading this, I think it might be helpful to have a box somewhere that includes simple definitions of the various relevant variables that the reader could refer to. This could include, for example: track segment, bout, song track, etc.

P 12, L 275-276. The first part of this sentence is unclear. What do the authors mean by "after the first 7.3 h of his sound bout compared to the following 8.7 h of the bout?"

P 12, L 279. Suggest change "at a" to "as his."

P 13, Discussion section. This section is still fairly lengthy, and dedicated mainly to the potential interpretation/implications of the data. A lot of new background information is presented - could any of this be included in the Background section?

P 13, L 297-298. The second sentence of the discussion doesn't really seem to follow from the first. It seems that the authors are trying to make the point that the males with the highest physical fitness (and thus those able to swim 23 km/h while singing) are the ones singing while swimming faster than the average whale? This could be presented more clearly. It also might be helpful if the authors spent a bit more time at the beginning of the discussion summarizing the main results, what they mean, and why they are valuable in and of themselves, rather than jumping right in to talk about the potential implications of the results and future (interesting!) research questions.

P 14, L 321-325. This section regarding future research directions is a great addition here.

P 15, L 338. Suggest change to "...and singing would provide the modality..."

P 17, L 387. Should read "...possess an equal..."

Decision letter (RSOS-180525.R1)

11-Feb-2019

Dear Dr Clark:

On behalf of the Editors, I am pleased to inform you that your Manuscript RSOS-180525.R1 entitled "Fin Whale Singing Decreases with Increased Swimming Speed" has been accepted for publication in Royal Society Open Science subject to minor revision in accordance with the referee suggestions. Please find the referees' comments at the end of this email.

The reviewers and Subject Editor have recommended publication, but also suggest some minor revisions to your manuscript. Therefore, I invite you to respond to the comments and revise your manuscript.

- Ethics statement

- Data accessibility

<http://datadryad.org/submit?journalID=RSOS&manu=RSOS-180525.R1>

- Competing interests

- Authors' contributions

- Acknowledgements

- Funding statement

Because the schedule for publication is very tight, it is a condition of publication that you submit the revised version of your manuscript before 20-Feb-2019. Please note that the revision deadline will expire at 00.00am on this date. If you do not think you will be able to meet this date please let me know immediately.

Kind regards,
Andrew Dunn

Royal Society Open Science Editorial Office
Royal Society Open Science
openscience@royalsociety.org

on behalf of Dr Ari Friedlaender (Associate Editor) and Kevin Padian (Subject Editor)
openscience@royalsociety.org

Associate Editor Comments to Author (Dr Ari Friedlaender):

To the Authors,

I applaud the authors for taking the considerable effort to incorporate the comments and suggestions of the reviewers. It is clear that this study is of interest, that the changes made to the analyses make it more scientifically sound, but there are still a couple of important revisions that are required before it can be accepted. As the reviewers point out, there is still some discrepancy between what is considered 'fast' swim speeds as they relate to singing and the reviewer provides an excellent assessment of the issue and how to consider it. Additionally, both reviewers (and myself) are still hesitant to sign off on the discussion of how a whale would perceive fitness from sound and inferring speed. The reviewers make an excellent point that given the amount of logistical support required to simply collect and evaluate this information, if/how a whale can do this is completely unknown. While the discussion does to a much nicer job of synthesizing the results, I completely concur with both reviewers that 'It would be a simpler more defensible discussion if the authors just noted that whales' ability to judge differences in speed through audition are unknown but must be acute for songs to provide useful stamina cues of the sort being proposed.'

Please see the full comments by reviewers that must be addressed before the manuscript can be accepted.

Thank you.

Ari S. Friedlaender

Reviewer comments to Author:

Reviewer: 2

Comments to the Author(s)

Re-Review of "Fin Whale Singing Decreases with Increased Swimming Speed"

In this revision, the authors have taken care to address the concerns of myself and the other original reviewer. As before, this work clearly demonstrates systematic differences in fin whale song production depending on swimming speed. The additions related to song duration strengthen the data set. The work is important and scientifically sound, and the analyses are appropriate.

The suggestion that listeners can assess male fitness based on swimming speed (with singing as the mechanism for determining speed) is interesting. While still speculative, the authors do a much better job of presenting their hypothesis as just that, and acknowledging their assumptions. Whether song production might also be an indicator of male quality remains to be seen, and the manuscript is simpler and more compelling without this additional assertion. I do think the discussion section could be shortened and made even simpler/more concise for clarity, with

more emphasis on the direct meaning of the results rather than just the interesting potential implications.

Based on the authors' revisions, I can now recommend this manuscript for publication.

A few specific comments follow:

P 8, L 166. Effect should be affect.

P 8, L 178-179. Why was a duty cycle >70% defined as "robust?" Was this an arbitrary distinction?

P 9, L 198-200. Should this information be provided in the Results rather than the Methods section?

P 9, L 208. Remove comma after "initially."

P 11, L 251. As I am reading this, I think it might be helpful to have a box somewhere that includes simple definitions of the various relevant variables that the reader could refer to. This could include, for example: track segment, bout, song track, etc.

P 12, L 275-276. The first part of this sentence is unclear. What do the authors mean by "after the first 7.3 h of his sound bout compared to the following 8.7 h of the bout?"

P 12, L 279. Suggest change "at a" to "as his."

P 13, Discussion section. This section is still fairly lengthy, and dedicated mainly to the potential interpretation/implications of the data. A lot of new background information is presented - could any of this be included in the Background section?

P 13, L 297-298. The second sentence of the discussion doesn't really seem to follow from the first.

It seems that the authors are trying to make the point that the males with the highest physical fitness (and thus those able to swim 23 km/h while singing) are the ones singing while swimming faster than the average whale? This could be presented more clearly. It also might be helpful if the authors spent a bit more time at the beginning of the discussion summarizing the main results, what they mean, and why they are valuable in and of themselves, rather than jumping right in to talk about the potential implications of the results and future (interesting!) research questions.

P 14, L 321-325. This section regarding future research directions is a great addition here.

P 15, L 338. Suggest change to "...and singing would provide the modality..."

P 17, L 387. Should read "...possess an equal..."

Reviewer: 1

Comments to the Author(s)

See attached file

Author's Response to Decision Letter for (RSOS-180525.R1)

See Appendix D.

Decision letter (RSOS-180525.R2)

07-May-2019

Dear Dr Clark,

I am pleased to inform you that your manuscript entitled "Fin Whale Singing Decreases with Increased Swimming Speed" is now accepted for publication in Royal Society Open Science.

on behalf of Dr Ari Friedlaender (Associate Editor) and Kevin Padian (Subject Editor)
openscience@royalsociety.org

Associate Editor Comments to Author (Dr Ari Friedlaender):

To the Editor,

I have reviewed the second revision of this submission and believe the authors have done considerable work to repackage the results of their work so they better align with the limitations and abilities of the study. I am confident that this interesting work will be of interest to a wide range of scientists. The value of this work is high and relates to a number of scientific practices. I laud the authors for their work and perseverance throughout this process.

Thank you.

Ari S. Friedlaender

Follow Royal Society Publishing on Twitter: [@RSocPublishing](https://twitter.com/RSocPublishing)

Appendix A

Clark and colleagues describe acoustic observations of singing fin whales that reveal systematic differences in song production that correlate with variations in how fast a whale is moving. The findings provide an important window into the behavior of singing whales. The methods used to analyze movements and sound production by singing fin whales are appropriate and the statistical summaries of measures are clearly presented. The findings reported are solid and important. The conclusions drawn by the authors are not, however, supported by their findings or any other evidence in the literature. For this reason, I cannot recommend that this paper be published in its current form.

The clear implication of their data is that continuously producing “robust songs” during fast swimming is either not possible or is maladaptive, because no whale swimming more than 12 km/hr sings with a duty cycle greater than 60%. This implies that singers have to choose between singing robustly or swimming fast. Presumably, a major factor controlling such choices would be whether a singer is doing fine where they are or have some motivation to be somewhere else. The authors do not mention why fin whales would ever bother to sing robustly while swimming slowly, if listeners are ultimately going to judge the singers based on what they do vocally while swimming rapidly. In fact, the authors suggest that singers should engage in “endurance displays” (sporadic singing while swimming more rapidly) as often as possible to impress the maximal number of listeners. Given that only 11 of 158 whales (7%) have an average duty cycle over 50% while swimming faster than 10 km/hr, it seems that most singers are not doing this. Instead, ~45% are swimming (or drifting) at less than 5km/hr with a duty cycle > 70%. Most fin whales seem to be choosing the swim-slow-sing-robustly option.

The authors’ conclusion that fin whales that intermittently sing while swimming more rapidly are attempting to “show off” to other whales is based on a number of unwarranted assumptions rather than on any evidence. This post-hoc explanation for the observed correlation serves mainly to make the authors’ findings fit current assumptions about why fin whales are singing. To my knowledge, there is no evidence of any other vertebrate rapidly running, swimming, flying, or traveling through any other means while intermittently loudly vocalizing to advertise their fitness. If there are any such reports, the authors should cite these.

There is no evidence that fin whales that attempt to sing while rapidly swimming are more reproductively fit than fin whales that don’t. One could just as easily assume that whales attempting to sing while rapidly swimming are younger fin whales that are not experienced enough to have learned that singing while rapidly swimming is a waste of time and energy. Or, perhaps whales only sing while rapidly swimming under special circumstances that are not the conditions most frequently encountered by singing fin whales. The only rationale given for favoring the possibility that “fast-swim singers” are uber-males is the assumption that there must be some way that listening whales are sorting singers based on their fitness, and that this is the only feature of fin whale singing that seems to vary across individuals in a way that the authors think could be used by listening whales to judge individuals. The fact that this possibility is conceivable provides no support for it.

The authors suggest that the capacity to sing while rapidly swimming might be a reliable indicator of fitness, but no measures of intra-individual variability of duty cycles produced during fast (or slow) swimming were presented in the report, so there is no way to assess whether any individual whale consistently sings at some maximum duty cycle while swimming fast. The presented data are consistent with the possibility that on some days singers will sing while fast swimming, but on other days they will not. The authors assume that whales that swim fast without singing would be singing while swimming if they were capable, but the authors present no evidence that the singers are not capable of doing so. If intra-individual variability of song production across days is comparable to inter-individual variability, then fast-swim singing would be useless as a fitness cue.

The idea that singing is energetically costly for baleen whales has been considered before in the literature, particularly in the case of singing humpback whales (e.g., Chu, 1988). Generally, there is no reason to believe that the energetic cost of singing is high. For instance, there is no evidence that dive times of singers are shorter than those of non-singers, which might be expected if singing were effortful. The fact that whales can sing almost continuously for periods of 24+ hours further suggests that singing does not require a large energy expenditure. Singing while swimming fast might be difficult for reasons unrelated to the amount of energy required. For example, the mechanics of sound production might be affected by rapid tail movements, or if a whale were swimming closer to the surface and blowing more frequently. The assumption that singing is energetically costly is not supported by any available data from baleen whales and is unnecessary to account for the observed behavioral correlation.

The authors assume that other whales are monitoring and tracking singers over periods of hours or more. It is unclear how long a listener would need to monitor a singer to be able to make a decision about the singer's capacity to sing while swimming fast. What is the probability that a listener would be in range of a singer long enough to make such a judgment? More generally, it is unclear that other whales are doing anything like this. The proposed scenario is highly speculative and assumes that listening fin whales possess several auditory and memory capacities that have never been shown in any species, including humans.

If the authors' preferred interpretation of their findings were presented as such in the conclusions, and noted as being a post-hoc envisioning of how the observed phenomenon might relate to song function, then it would not be problematic. The current presentation of the observations instead is likely to give many readers the misimpression that the data provide evidence supportive of the authors' interpretation. For example, in the abstract, the authors state that, "The attributes of male acoustic advertisement displays are often related to a performer's age, breeding condition and motivation. We examined this relationship in fin whales, measuring a singer's swimming speed and amount of singing." This is not an accurate portrayal of the analysis that was performed, as it did not relate to age, breeding condition, motivation, or any other feature of an acoustic display that has been studied in any other acoustically displaying animal. The second sentence implies that swimming speed and singing duty cycle are known to be reproductively relevant attributes of songs, which is not the case.

Later in the abstract, the authors state that, “Given the high intensity of song notes and long durations of song bouts, we assume that singing is energetically costly. This suggests that the combination of swimming speed and singing rate is an endurance display by which females and/or other males can assess a singer’s physical fitness and potential reproductive quality.” Even if singing is energetically costly, which has yet to be established, this would not imply that fast-swim singing is an endurance display, because there is no evidence that fast-swim singing is difficult over the time spans that singer are doing it, or that it requires any more endurance than robust singing. This description presents a series of assumptions about a possible scenario with no empirical support as if it directly followed from the observations that the authors report.

I believe that the paper would be greatly improved by presenting the findings in a way that emphasizes the last point made in the abstract that, “Results have implications for interpreting fin whale singing behavior and the possible influences of anthropogenic sounds on fin whale mating strategies and breeding success.” The main significance of the observations lies in what they reveal about what fin whales are doing when they’re singing. Given the durations of the reported singing bouts, it seems likely that produced songs are effective regardless of a singer’s average swimming speed.

The example of non-robust song shown in Figure 2, if representative, suggests that non-robust singing is more temporally irregular, and Figure 3 seems to suggest that these temporal irregularities might relate to changes in swimming speed. This finding seems similar to earlier reports that irregular structure in humpback whale songs is related to variability in dive times (Chu & Harcourt, 1986). It seems likely that the dive pattern of the singer shown in Figure 3 would be less regular than that of the singer shown in Figure 1. The authors cannot tell for sure when the singers they are tracking were surfacing, but they could still speak to the possibility that decreased dive times or less regular dive times might be correlated with changes in swimming speed, and the possibility that variations in dive duration might partly compensate for any increases in energy expenditure and/or account for intermittency in singing bouts.

Insufficient data is shared to support the article results. I assume this is because the data is from the SOSUS array. It would be preferable if the authors could at least provide audio files for the samples that they show in Figures 1-4.

Appendix B

Review of “Fin Whale Singing Decreases with Increased Swimming Speed”; Clark, Gagnon and Frankel, Royal Society Open Science

General Comments

Reviewer-1: This manuscript describes an apparent relationship between swimming speed and duty-cycle for singing fin whales. It is well written, with a variety of relevant background information provided. This work is interesting given the difficulty of studying breeding behavior in free-ranging fin whales, and inspires a range of additional questions worthy of future research.

The main issue with this manuscript is that the authors must be careful not to assert their speculations about honest signaling as demonstrated fact. While they have clearly shown that there is a correlation between swimming speed and singing, they have not demonstrated that this is an “endurance display” that is behaviorally meaningful to females or other males. The authors raise many interesting questions that may inspire new research, but it must be made clearer throughout that additional research is needed to confirm (or refute) their claims. While the work itself is sound, before this paper is suitable for publication the authors need to adjust the scope of the manuscript so as not to overextend the implications of their results.

Reviewer-2: Clark and colleagues describe acoustic observations of singing fin whales that reveal systematic differences in song production that correlate with variations in how fast a whale is moving. The findings provide an important window into the behavior of singing whales. The methods used to analyze movements and sound production by singing fin whales are appropriate and the statistical summaries of measures are clearly presented. The findings reported are solid and important. The conclusions drawn by the authors are not, however, supported by their findings or any other evidence in the literature. For this reason, I cannot recommend that this paper be published in its current form.

Authors Responses to General Comments by Reviewer-1 and Reviewer-2:

There were many excellent reviewer comments that challenged our earlier thinking and analysis. We took these seriously. As a result, we devoted considerable time and effort to rebuild and revise the data set of acoustic observations. As a result of this major effort, we expanded the data to include all months of the year (not just Aug – April), nearly doubled the number of acoustic locations from 609 to 1208 and added a second metric on song production by including data on 8023 song durations for the 163 singers. The data on song durations shows an inverse relationship between duration and swimming

speed, which is consistent with an early hypothesis by Chu and Harcourt (1986) suggesting that in humpbacks, song duration could serve as an indication of stamina and physical condition. Our analysis results reveal that both song production metrics, duty-cycle and song duration, are inversely and significantly correlated with a singer's swimming speed. The faster a male singer swims the less time he sings and the shorter his songs.

We used these comments to look seriously at our thesis and results. In particular were those comments that pointed out when our statements and assumptions were not supported and that some of those assumptions were built on previous assumptions. In that reviewed manuscript, we had interpreted our observations of fin whale swimming speed and amount of singing (the duty-cycle [DC] metric) as evidence that the energetic cost of both swimming and singing combined was the energetically demanding behavior and therefore the combination of high swimming speed and high duty-cycle was the means by which listening conspecifics, both female and male, could assess a singer's stamina and thus male quality. As pointed out in the reviews, we can only assume, but have no evidence that singing is energetically costly. Therefore, in this revised version, we have simplified our interpretation of the data showing an inverse relationship between swimming speed and amount of singing such that we now assume that swimming speed alone, not swimming-singing combined, is a direct indicator of energetic cost. Although there is empirical evidence that increases in vocal effort can result in increases in metabolic rate in at least one cetacean (Holt et al. 2015), we assume that the energetic cost of swimming is far greater than the cost of singing. This assumption is supported by empirical data for other species and by model results for baleen whales (Hind et al. 1997, Braithwaite et al. 2015). We then further propose that because swimming speed is energetically costly (energetic cost increases with speed to the third power), swimming speed can serve as an indication of male quality, while singing is the modality by which listeners assess a singer's swimming speed. Whether or not singing while swimming imposes an energetic cost that compromises song production, such that measures of song production can serve as reliable proxies for male quality, remains an interesting question that we cannot answer with our present data.

Based on the improvements in our data set, additional GAM analysis and in consideration of the reviewers' comments, we have revised the text throughout the document, especially text related to our earlier unsupportable assumptions about the cost of singing and the conclusions about singer stamina and singer quality.

Most fin whales seem to be choosing the swim-slow-sing-robustly option.

Specific Comments from Reviewer-1 with Authors responses: (Note that Reviewer-1 used page and

line numbers in the reviewed manuscript. The authors' response uses page and line numbers in the revised manuscript.)

P. 1, L. 15. Suggest "...improving **understanding of** the potential effects..."

Response, P. 1, L. 16: Accepted and changed

P. 1, L. 16-17. Are the singers mentioned here known or presumed males?

Response, P. 1, L. 16-17: We presume singers are males and have added the phrase "...**singers, which are presumed to be males, swam...**". This presumption is based on the study referenced in Croll et al. (2002) during which we tagged, biopsied and acoustically tracked singers and non-singers, and the biopsies revealed that all singers were males. Humpback and blue whale singers have all been males as well.

P. 2, L. 37. Cetacean should read **cetaceans**.

Response, P. 2, L. 38: Accepted and changed

P. 2, L. 34-38. This is true, but performance attributes really need to be linked to behavior or reproductive success to determine whether they are meaningful honest signals.

Response, P. 2, L. 38-39: We modified the ending phrase in the sentence to read "..., we **assume** that insights into possible performance attributes can be deduced from empirical observations."

P. 3, L. 50-52. Are there any examples of this in better studied animals? That would strengthen the authors' argument.

Response: This text has been removed. Throughout the manuscript, we have significantly changed the text so as to simplify the interpretation of our data and results. We have a) removed text that assumes or implies that the combined energetic cost of swimming and singing is the primary factor driving singer performance, b) focused instead on the simpler assumption that energetic cost of just swimming is the primary factor driving singer performance, and c) written that a listening conspecific can only assess a singer's swimming speed if the singer continues to sing. So, the

indicator of physical fitness/stamina is singer swimming speed, not swimming speed and singing, but singing provides the mechanism by which listeners can assess a singer's speed and thus possibly his physical stamina." We contend that this simplified interpretation of our results and the lack of examples in other taxa demonstrating that the ability of a male to sing while moving (running, flying, swimming etc.) fast is an indicator of singer physical fitness, negates the need to include references to such examples.

P. 3, L. 54. Effects should be **affects**.

Response: Corrected, thank you!

P. 3, L. 57-60. The authors are presenting an interesting hypothesis here, but it's one that needs further testing. There are several sentences in a row here with implications building on implications, and the authors should be careful not to overextend the results of their empirical finding.

Response: We agree with this comment. Throughout the manuscript (see above), we have significantly changed the text so as to simplify the interpretation of our data and results. The motivation for doing so was removed and avoided such over interpretation and extension of our results. In our revised discussion we do postulate (see lines 337-352) that: "Although the specific energetic cost of song production in a baleen whale is unknown, for fin whales it may be a factor in a full energy assessment of the acoustic courtship display. It is known that increased vocal effort in bottlenose dolphins produces increases in their metabolic rate, confirming that there is a cost to vocalizing.

P. 4, L. 72. Consist should be **consists**.

Response: Corrected, thank you!

P. 4, L. 89-92. Yes, it is *possible* that the quality of song performance under higher swimming speeds contains information about male quality. However, whether or not this is true still needs to be addressed with further research. If this is found to be the case, the next relevant question becomes: is this meaningful to females?

Response: We agree with this concern and have reduced the over speculation in the text while also

expanding the text to reiterate our observed associations between swimming speed and singing, the fact that swimming faster is energetically costly (see Hind et al. 1997) and to raise a number of questions as a result of our observations and results.

That paragraph now reads: “These observations peaked our interest and suggested that there was some reliable relationship between a male’s swimming speed and his singing behavior. Why would some males continue singing when they swam faster, and others would not? Was there something about this behavior that was somehow adaptive and reliably indicative of a singer’s physical stamina; something by which females could choose a higher quality mate and/or something by which competing males could decide whether or not to compete with or retreat from a singer, or instead were we observing a behavior that served little to no benefit to the singer? Regardless of the eventual explanation, we knew that the ability of a singer to sing, even if only intermittent song, while swimming at higher speeds provided a direct mechanism by which we could track a singer’s speed, and therefore, we assumed that this was also the mechanism by which listening whales could assess a singer’s speed. Given that energetic cost in whales has been shown to be directly proportional to swimming speed [31], supporting the assumption that swimming fast is a physically demanding behavior, this suggests that singing while swimming fast is a means by which conspecifics can possibly assess a singer’s stamina.”

P. 5, L. 96. Suggest “...higher speeds while continuing to sing **has the potential to be** an honest signal...”. Just because something could be an honest signal does not mean that it is biologically relevant or meaningful to other individuals in the population (or that it is actually utilized as a cue by conspecifics).

Response: Corrected, thank you. This now reads: “These observations are consistent with the hypothesis, but not enough to conclude, that the ability of a singer to swim at higher speeds while continuing to sing has the potential to be an honest signal indicative of the singer’s physical stamina and possibly his reproductive quality.”

P. 5. L. 108-109. It would be useful to provide some information about typical numbers of concurrent signers, how far apart they were, how long they were tracked on average, etc.

Response: As stated at the beginning of the methods section, for system security reasons we cannot give details about positioning and tracking resolutions. We do provide details about track lengths and durations and concurrent singers in the results section.

P. 5, L. 122. Suggest add comma after “through.”

Response: Sentence now reads “We avoided periods when ambient noise conditions were elevated as a result of ocean noise and occasions when a seismic airgun survey was operating within or a surface vessel was transiting through the area in which a whale was singing.”

P. 6, L. 117. What is this “very high confidence” based on? The next sentence provides some clarification on this, but more details would be helpful.

Response: We have rewritten this section so as to more clearly explain the observations. This confidence is a combination of the exceptional capabilities of the system to track low-frequency acoustic objects through space and time as well as our being extremely diligent and careful not to include a track if there was any uncertainty that it could be confused with the track of another singer. We have modified the text in this section to better emphasize how and why “we have very high confidence that there was no ambiguity in the assignments of track locations and all locations in a track represent the movement of the same singer and are not mixtures of two or more singers.”

The text in this section (lines 130-145) now reads: “We used the time-varying features of each singer’s song track (i.e. speed and direction of travel), as well as the acoustic features observable in visual displays (e.g. spectrograms) containing each singer’s songs, to unambiguously distinguish between different singers and tracks. Tracks also included information on the distance, bearing and speed between locations throughout an individual’s track. All song tracks were mapped and carefully reviewed. If there was any ambiguity as to whether or not a track was derived from the same singer, that track was not included in further analysis. Thus, we have very high confidence that there was no ambiguity in the assignments of track locations to a single individual and that all locations in a track represent the movement of the same singer and are not mixtures of two or more singers. We are also very confident that the track data for each individual singer cover the entire period of the singer’s song bout(s). We assume that the same singer was not tracked more than once in the same year, but we cannot rule out this possibility, especially for the multi-year data. This process, which was designed to eliminate possibilities of confounding multiple tracks, does introduce some unknown level of bias against including tracks during periods of high singer density.”

P. 6, L. 121-122. How far apart did singers have to be in space and time to be considered separated from one another?

Response: We did not set minimum distance or time thresholds to distinguish between tracks. Because of the systems' analytical attributes time was not a critical factor in distinguishing tracks. We did add text to this section: "Successive locations in a track were at least 10 km apart, but we did not apply a minimum separation distance cut-off as the basis by which to decide if a location was part of the same track."

P. 6, L. 124-127. What do we know about patterns of swimming for this species generally? Is it likely to vary a lot over time (in which case an average across locations may obscure important details)? What about behavioral context?

Response: Swimming patterns in this species are poorly understood and almost exclusively based on data from either satellite tags, radio tags or more recently a few archival tags. Satellite tag data typically provide a few coarse positions throughout a day and coarse patterns of swimming can last for many months, but do not provide acoustic data. Radio tags, like most satellite tags, can last many weeks and be used to follow individual animals (e.g. Watkins 1981).

We added text lines 73-77: "Fin whale swimming speeds have been measured for singers and tagged whales. Singers in the North Pacific have been acoustically tracked while swimming between 1 and 14 km/h [24, 29]. Radio-tracked fin whales of unknown sex and unknown acoustic activity have been observed swimming between 10-16 km/h for days, and achieved speeds of 20 km/h or greater for at least 20 min [30]."

The sex of the satellite or radio tagged animal is not typically known, and these tags did not collect acoustic data, so results from these tag studies are very limited for informing patterns of swimming for singing males. Context from radio tags is interpreted based on visual observation from a following vessel (i.e. focal follows), such that observers can distinguish between foraging, traveling and socializing behaviors. These studies do have not taken place during the winter breeding season when singing predominates. In a few cases, the observers deployed hydrophones or a small array to monitor if sounds were detected and possibly associated with the tagged animal. Because of the logistical difficulties of tag deployment, focal follow studies have been rare. Archival acoustic tags are typically attached with suction cups, provide exceptional details

on the 3D movements of the tagged animal, can last for most of a day, and provide excellent acoustic data on an individual. As with other tags, such studies rarely occur in the winter breeding season when singing is prolific. The limitations of these previous studies using tag technologies within the context of the male acoustic display, underscore the uniqueness of our study's data, collected throughout an enormous ocean area, throughout the year, over multiple years, using an exceptionally sophisticated passive acoustic system. The data presented here are solely based on an extraordinary passive acoustic detection-location-tracking system that operates continuously and covers vast areas of the North Atlantic Ocean.

We are in total agreement with reviewer comments concerning context, and we have added text in several places to clarify that we do not know anything directly as to the behavioral-ecological context for the acoustically tracked whales. For example, see lines 310-320: "Because we only have information about swimming and singing behavior, and no direct knowledge of the actual behavioral context or how listening whales might have responded, we have no means by which to address these questions with our present results. Because our techniques are best used to address questions related to male-male interactions, in the future we could specifically focus on places and time periods when we have multiple singers within relatively close proximity. Under these conditions we might expect some singers to approach and some to avoid other singers as evidenced by their tracks and swimming speeds, and these movement dynamics should be accompanied by changes in singing behavior. Although it requires considerable effort to track multiple singers within close proximity, we could test whether or not singing behaviors are only related to speed or somehow related to this context."

Thus, as we state, we can only infer context based on existing information that a) all known singers for which sex has been determined have been males and b) singing is prolific both over space and time (see ref Clark and Gagnon 2004).

P. 6, L. 136. Are the authors confident that all singing was captured? What proportion of vocalizations were thought to be recorded?

Response: Yes, given the system's capabilities that were designed to detect, locate and track relatively quiet acoustic objects, we are very confident that all our tracks include the complete time period for a singer's song bout(s) and are no tracks are a combination of multiple singers. We have included text to further clarify and emphasize our confidence that all data for a singer are included in each track (lines 140-145): "We are also very confident that the track data for

each individual singer cover the entire period of the singer's song bout(s). We assume that the same singer was not tracked more than once in the same year, but we cannot rule out this possibility, especially for the multi-year data. This process, which was designed to eliminate possibilities of confounding multiple tracks, does introduce some unknown level of bias against including tracks during periods of high singer density."

P. 6, L. 136. How does ambient noise affect the range over which these calls are detectable by the hydrophone arrays?

Response: We have added text to clarify that we did not include tracks of singers under high natural ambient noise conditions that typically occurred as a result of ocean storms. Lines 125-127: "We avoided periods when ambient noise conditions were elevated as a result of ocean noise and occasions when a seismic airgun survey was operating within or a surface vessel was transiting through the area in which a whale was singing."

P. 7, L. 142. What about variation in song duty-cycle and whale speed within a track? How was this accounted for?

Response: We have included all singer summary statistics for all 163 individual tracks as Supplementary Materials (Table_S1). We did not include variation in speed, duty-cycle or song duration in our GAM models. We did compare coefficients of variation (CV) for duty-cycle, and song duration as a function of swimming speed and found no differences: that is, variations in singing (duty-cycle and song duration) did not depend on swimming speed such that, for example slower swimming singers had lower variation in song features than faster swimming singers.

P. 7, L. 153. How far apart were these locations typically? Was there a maximum distance allowed between two points to be considered the same track?

Response: We cannot divulge the precise answer to this first question, because that could reveal classified information about the system performance, but we have added the following text at lines 214-217: "Successive locations in a track were at least 10 km apart, but we did not apply a minimum or maximum separation distance cut-off as the basis by which to decide if a location was part of the same track."

P. 7, L. 161-163. Are these values across the full tracks? Across two locations? Again, what was the variability like within a track?

Response: These values were for track distances, not track segment distances; where a track segment distance is the distance between two sequential location. We have rewritten this and all results sections to include more track, swimming speed and song metric statistics. For example, Lines 247-255: “Track segment speeds for all singers varied from < 1 to 23 km/h ($n = 1208$), and the mean segment speed for all singers was 6.4 km/h (s.d. = 3.4, median = 6.2). The number of segments per track ranged from 2-38, and the average number of segments per track was 7.4 (s.d. = 7.1, median = 5). The full dataset of segment speeds was used to describe the frequency of occurrence of every combination of duty-cycle and swimming speed (figure 3). The distribution of track segment speeds showed a strong bimodal duty-cycle distribution: one is a high duty-cycle mode (75-90% duty-cycle range), with speeds distributed between 1-8 km/h. The second is the low duty-cycle mode 0-5% duty-cycle range, with speeds distributed primarily between 1-10 km/h.” This section of the results has been rewritten to include more statistics on track

P. 8, L. 167. What does “some singers” refer to? What proportion of singers exhibited this behavior? How long were they able to sustain this behavior?

Response: We recognize that the form of words “some singers” was not very informative and that we needed to include more information in the results. We have essentially rewritten all results sections and removed vague forms of words such as “some singers.” An important part of the results section rewrite involved looking at the distribution of swimming speeds, the speed at which a singer started his bout and how the singer changed his speed relative to the population average swimming speed throughout his singing bout. This section now reads (lines 237-246): “Most of the 163 singers started their song bouts at speeds < 7 km/h (57%). Most of these either continued swimming at < 7 km/h (55%), switched to swimming faster and continued swimming faster for the remainder of their track (12%), or switched back and forth between swimming faster and swimming slower during the track (33%). For the singers that started their song bouts swimming at ≥ 7 km/h (43%), most either continued swimming at ≥ 7 km/h (50%), switched to swimming slower and continued swimming slower for the remainder of their track (17%), or switched back and forth between swimming slower and swimming faster during the track (33%). Although the majority of singers started their song tracks swimming at < 7 km/h, only 31% of all

singers continued swimming at < 7 km/h for their entire song bout(s). In contrast, 69% of all singers swam at speeds ≥ 7 km/h during some period of their track.”

P. 8, L. 170. Might song length be an alternative honest signal?

Response: Yes (Thank you for bringing this up!), song duration (“length”) is another possible indicator of singer quality (see Chu and Harcourt 1986). We made a great effort to go back into the data archives to extract song durations ($n = 8023$) for all 163 singers, and we added song duration as a factor in GAM analysis. Along with appropriate additions to the text, the results now include totally new Figures 6, 7 and 8.

P. 8, L. 171. Why are the faster-swimming singers swimming faster? Are they really stopping singing because of energetic costs or because they are switching behaviors for some other reason? Context matters here, and these details are important to consider.

Response: These are really good questions, and questions that cannot be answered from a passive acoustic method alone or from our present data. We have rewritten discussion text to do our best to recognize the limitation of what we can interpret from our observations. We have also attended to reviewer comments regarding over-interpretation by qualifying suspicions or whenever we attempt to draw conclude from our results. One important component of this revision is that we have removed amount of singing (e.g. duty-cycle or song duration) as an energetic consideration and instead only invoked swimming speed as having an energetic cost. This simplifies the discussion, while recognizing that everything we present is qualified by the fact that we do not know the behavioral-ecological contexts other than time of year within the North Atlantic Ocean, and the most likely assumption that this is a male acoustic reproductive display.

P. 8, L. 176. Suggest “...158 presumed unique singers...”

Response: This specific text has been deleted and replaced in the results section.

P. 8 – 9, L. 181-191. These examples are very useful. Can the authors comment on intra-individual variability in these patterns over time/season/context?

Response: We do not know behavioral context, except that we recorded and tracked singers

throughout the year and we assume, based on direct evidence from Croll et al. (2002), that these are males. We have added more analysis details in the results section (Lines 207 -273), including an overview of how individual singers changed their swimming-singing behavior throughout the track. This revealed that there were two modes in the distribution of duty-cycle (see figure 3) and lines 248-253: “The full dataset of segment speeds was used to describe the frequency of occurrence of every combination of duty-cycle and swimming speed (figure 3). The distribution of track segment speeds showed a strong bimodal duty-cycle distribution: one is a high duty-cycle mode (75-90% duty-cycle range, 34% of track segments), with speeds distributed between 1-8 km/h. The second is the low duty-cycle mode (0-5% duty-cycle range, 21% of track segments), with speeds distributed primarily between 1-10 km/h.” These results are a good indication that swimming fast while reducing amount of singing is not an aberrant or rare behavior.

P. 9, L. 200-208. These are great questions. Additional data are needed to address them! The data presented herein do not allow the authors to answer any of these questions, although it is helpful to discuss them and it would be great if the authors suggested some additional targeted research.

Response: We totally agree and now state, alas, that our data do not allow answers, only suggestive glimpses into what might be driving these observations. As we write on lines 310-317: “Because our techniques are best used to address questions related to male-male interactions, in the future we could specifically focus on places and time periods when we have multiple singers within relatively close proximity. Under these conditions we might expect some singers to approach and some to avoid other singers as evidenced by their tracks and swimming speeds, and these movement dynamics should be accompanied by changes in singing behavior. Although it requires considerable effort to track multiple singers within close proximity, we could test whether or not singing behaviors are only related to speed or somehow related to this context. Clearly, the best way forward would be to apply a multi-modal research approach involving, for example, acoustics, biopsy, photo-ID, tags (see Croll et al. 2002), and emerging techniques (e.g. Geoghegan et al. 2018).”

P. 10, L. 241-243. This is *not* implied. Just because information is available does not mean that it can be reliably used by other individuals.

Response: This is one of several comments regarding our unsubstantiated assumptions that were

often cascaded into a series of multiple assumptions. We have strived to remove such text and qualify when it contains assumptions. As mentioned earlier in this response-to-reviewers, we have changed our analysis approach so only assume that swimming is energetically costly (with support from references), and not assume that singing, nor the combination of swimming-singing, is energetically costly. This greatly simplifies our interpretation, while identifying the importance of understanding the basis for our observation that amount of singing (duty-cycle and song duration) is inversely related to swimming speed.

P. 10, L. 248. A listening individual is just one receiver. We don't know about the localization abilities of this species or how good a fin whale would be at evaluating the location and speed of another singer. The authors are making large assumptions about what is known and what is possible – these caveats must be acknowledged.

Response: These are valid points, and we have taken care in the revision to not overstate assumptions, nor to try to argue that fin whales in particular or baleen whales in general have certain acoustic capabilities. We have added text with references (see lines 356-372) to expand on how a listening whale might evaluate and gain information about a singer's swimming velocity (bearing and range).

P. 10, L. 251-257. Do we have evidence that whales can actually do this?

Response: We do not have direct evidence that whales can do what we describe. We have added a paragraph that includes direct evidence via playback experiments that whales can locate sound sources and includes results from the application of sound transmission analysis indicating that range and bearings to whales acoustically active whales can be calculated. See also response to last comment.

P. 10, L. 244-259. There are lots of complicated, interacting factors presented here, and lots of assumptions. Again, the authors must be careful not to overextend their results.

Response: We have taken these concerns seriously and rewritten much of the offending text. See also responses to last two comments.

P. 12, L. 274. What is this range based on? Is this under "typical" ambient noise conditions? How might

auditory capabilities influence this?

Response: We have modified the text (now at Lines 397-401): “We conservatively estimate that under typical ambient noise conditions a listener could hear and follow the movements of a singing male within a listening range of 10-50 km, while acknowledging there would be considerable variability in this range depending on ambient noise and sound transmission loss conditions in the 18-28 Hz frequency band, and the depths of the singing and listening whales.”

Regarding whether auditory capabilities would influence detection range: The auditory capabilities for baleen whales have been inferred from anatomical models from a few baleen inner ears, which show that mechanistically, those ears are adapted for low-frequency hearing (see Ketten 1997). We did not want to delve into speculation about baleen whale audition as we did not feel it was a necessary component of this paper.

P. 13, L. 284. Why are receptive females likely the most limiting resources for males? What is this based on? Do we know that female preference is the predominant driver of male behavior in fin whales?

Response: We have added text and a reference (Agler et al. 1993) stating that: the “mean calving interval in fin whales is greater than two years” as support for females being a limiting resource.

P. 13, L. 294-297. This would be useful to mention or include a bit earlier (when listening distances are being discussed).

Response: We have decided to leave the text as is on this matter of distances over which listeners might hear singers.

P. 13, L. 298-299. This may be true for this study, but this could be confirmed other ways (visual surveys, tagging, etc.).

Response: We have added the phrase “using this passive listening array system” to the first sentence of this paragraph (now lines 417-418).

P. 14, L. 312-313, 322-326. These are great points! The authors do a good job here of addressing some

caveats of their assertions, but still need to be more explicit about this.

Response: We have modified the text in this section based on an analysis of individual swimming tracks so that it now reads (lines 440-451) “A possible explanation for why some singers stopped singing as their swimming speeds increased is that they actively switched to another behavior, as occurs in humpback whale males [69]. We cannot rule this out, but this does not explain why the majority of swimming singers ($n = 112$, 69%) sang at swimming speeds >7 km/h, while less than a third of singers ($n = 51$, 31%) only sang at swimming speeds <7 km/h. If so, many singers are successful in finding or attracting a female at higher swimming speeds, then why do nearly a third of males only swim slower while singing? This would lead to a conclusion that nearly a third of males adopt a behavior that is less physically demanding and not likely to lead to a benefit greater than gained by swimming fast. Given that swimming fast is energetically costly and possibly an indication of a male’s physical stamina, this suggests that our observation of a highly significant inverse relationship between swimming speed and singing is consistent with the conclusion, although speculative, that swimming speed is an honest signal by which conspecifics can assess male quality.”

P. 14, L. 317-318. Maybe there are other reasons for this! Animals do not always perform optimally.

Response: see last response as we have revised this paragraph based on an analysis of individual swimming tracks.

P. 14, L. 321. This is still speculative, and cannot be concluded directly from the observations/analysis in the present study.

Response: We agree and have inserted the words “although speculative” into the text.

P. 15, L. 347. Suggest “...ability of listeners to **detect or** assess the quality...”

Response: This section of text has been deleted.

P. 15, L. 351. Suggest “...duty-cycle might explain **one way in which** females and/or other males...”

Response: This suggestion was accepted so text reads (line 469): “For fin whales, the observed relationship between male swimming speed and amount of singing might be one way by which

females and/or other males assess the quality of a male singer.”

P. 16, L. 355. This has not really been discussed in this way (i.e. that singing has a relatively lower energetic cost) previously in this paper, is this an important point?

Response: As stated in several earlier responses, we have abandoned the notion that singing imposes an energetic cost, and this revises our discussion on the swimming-singing relationship.

P. 16, L. 364. It would be useful to suggest future research that could confirm or refute the hypothesis presented herein.

Response: We have included some suggestions earlier in the text. In reality, the task of confirming or refuting the hypothesis that swimming fast is an honest indicator of male stamina and quality seems daunting given that acoustic experiments with large baleen whales are exceptionally difficult both logistically and legally (They are endangered). Thus, we would contend that under present conditions, the most likely way of advancing the science would come through multi-modal, field research and large-scale observations, such as those reported here. That said, once-upon a time, which is not that long ago, the recent baleen whale research projects would have seemed impossible!

P. 21, Figures 1 and 2. Suggest noting duration of x-axis in the caption in each case so the reader can more easily see how to compare between the figures.

Response: Thank you, we have amended caption to note x-axis duration.

P. 21, Figures 3 and 4. Suggest adding (left y-axis) and (right y-axis) to the caption to denote swimming speed and duty-cycle.

Response: Thank you, we have amended caption to note (left y-axis) associated with Speed and (right y-axis) associated with Duty-cycle. Please also note that previous Figures 3 and 4 are now Figures 4 and 5.

P. 21, Figures 3 and 4. These examples are helpful!

Response: thank you.

Specific Comments from Reviewer-2 with Authors' responses:

1. Reviewer-2, Item 1: Clark and colleagues describe acoustic observations of singing fin whales that reveal systematic differences in song production that correlate with variations in how fast a whale is moving. The findings provide an important window into the behavior of singing whales. The methods used to analyze movements and sound production by singing fin whales are appropriate and the statistical summaries of measures are clearly presented. The findings reported are solid and important. **The conclusions drawn by the authors are not, however, supported by their findings or any other evidence in the literature.** For this reason, I cannot recommend that this paper be published in its current form.

Response: Upon careful consideration of this review, we realized that this reviewer was correct when stating that our conclusions were not necessarily supported by our findings or the evidence as presented via our references. The following is how we have addressed these problems.

- a. We agree that we cannot justify one of our primary, initial assumptions that singing is energetically costly (e.g. Reviewer-2 comments items 2, 5, 6). By coupling this unjustified assumption about the energetic cost of singing with the cost of swimming speed, we weakened our proposed conclusion that singing while swimming fast was an indication of stamina, male quality etc. In this revision, we have eliminated any assumption about the cost of singing and only consider swimming speed as the behavior linked to energetic cost and thus possibly to stamina, male quality etc. Given the evidence that the energetic cost of movement in marine mammals (Davis et al. 1985), including a cetacean (Gallagher et al. 2018), is a function of speed to the third power, we believe our assumption that the energetic cost of swimming in fin whales is supported, and we now only consider swimming speed as the behavior linked to energetic cost and thus possibly to stamina, male quality etc.
- b. With considerable effort, we rebuilt our data set. We also added new analyses and modified our interpretations of the results. The expanded data set now includes:
 - i. 163 tracks from throughout the year, whereas before we had 152 tracks from 9 months, August – April, of the year),

- ii. Swimming speeds and associated duty-cycle values for each of the 1208 track segments, whereas before we only considered average swimming speed and average duty-cycle values for each of the 163 tracks, and
 - c. Because of reviewer comments (e.g. paragraphs 2, 5, 6), we have added new analyses and results.
 - i. To address intra-individual variability questions and concerns, we drilled down into swimming speed and duty-cycle data for track segments tracks ($n=1208$). This revealed a strong bimodal duty-cycle distribution: a high duty-cycle mode (75-90% duty-cycle range, 34% of all track segments), with speeds distributed between 1-8 km/h, and a low duty-cycle mode (0-5% duty-cycle range, 21% of track segments), with speeds distributed primarily between 1-10 km/h. The manuscript now includes Figure 3, which shows the 3D distribution of duty-cycle and swimming speed values for all track segments.
 - ii. The GAM analysis included number of songs and day-of-year as additional predictor variables. This revealed that song duration, as with duty-cycle, decreased significantly as swimming speed increased, that number of songs was not a significant predictor and that day-of-year was a significant predictor. A significant yearly effect was also detected.
 - iii. Over two-thirds of all singers that swam faster than the mean track swimming speed (6.7 km/h) continued to sing at higher speeds, some as high as 23 km/h.
 - d. We have modified our interpretations of the results.
 - i. Our results for fin whales are consistent with the hypothesis for humpback whales that song duration could be a measure of male singer quality and directly related to a singer's physical condition (Chu and Harcourt 1986)
 - ii. We focus on swimming speed alone as the basis for energetic cost. New figure 3 shows the duty-cycle value and swimming speed combinations for all track segments ($n=1208$).
- 2. Reviewer-2, Item 2: The clear implication of their data is that continuously producing “robust songs” during fast swimming is either not possible or is maladaptive, because no whale swimming more than 12 km/hr sings with a duty cycle greater than 60%. This implies that singers have to choose between singing robustly or swimming fast. Presumably, a major factor controlling such choices

would be whether a singer is doing fine where they are or have some motivation to be somewhere else. The authors do not mention why fin whales would ever bother to sing robustly while swimming slowly, if listeners are ultimately going to judge the singers based on what they do vocally while swimming rapidly. In fact, the authors suggest that singers should engage in “endurance displays” (sporadic singing while swimming more rapidly) as often as possible to impress the maximal number of listeners. Given that only 11 of 158 whales (7%) have an average duty cycle over 50% while swimming faster than 10 km/h, it seems that most singers are not doing this. Instead, ~45% are swimming (or drifting) at less than 5km/h with a duty cycle > 70%.

Response: See our more detailed responses to Item 1 above and Item 5 below. In this revision, we have eliminated any assumption about the cost of singing and only consider swimming speed as the behavior linked to energetic cost and thus possibly to stamina, male quality etc. Given the evidence that the energetic cost of movement in marine mammals (Davis et al., 1985), including a cetacean (Gallagher et al. 2018), is a function of speed to the third power, we believe our assumption that the energetic cost of swimming in fin whales is supported, and we now only consider swimming speed as the behavior linked to energetic cost and thus possibly to stamina, male quality etc..

3. Reviewer-2, Item 3: The authors’ conclusion that fin whales that intermittently sing while swimming more rapidly are attempting to “show off” to other whales is based on a number of unwarranted assumptions rather than on any evidence. This post-hoc explanation for the observed correlation serves mainly to make the authors’ findings fit current assumptions about why fin whales are singing. To my knowledge, there is no evidence of any other vertebrate rapidly running, swimming, flying, or traveling through any other means while intermittently loudly vocalizing to advertise their fitness. If there are any such reports, the authors should cite these.

Response: See our responses to Item 1 above. We have avoided weak and/or highly speculative assumptions and removed our conclusion that implies that singers that swim faster are showing off. Our intent in the revised version was to remove unsupportable assumptions, and we do not think that we are simply trying to fit our observations into “current assumptions about why fin whales are singing.” The basically simple assumption that singing is a male reproductive display to either attract females to mate, compete with male rivals for access to receptive females, or some combination of both remains the same. As illustrated in Figures 7 and 8, and in previous publications (Clark and Gagnon 2004), fin whales sing throughout more than half the year, and songs are extremely loud. Singing is not a trivial exercise.

4. Reviewer-2, Item 4: There is no evidence that fin whales that attempt to sing while rapidly swimming are more reproductively fit than fin whales that don't. One could just as easily assume that whales attempting to sing while rapidly swimming are younger fin whales that are not experienced enough to have learned that singing while rapidly swimming is a waste of time and energy. Or, perhaps whales only sing while rapidly swimming under special circumstances that are not the conditions most frequently encountered by singing fin whales. The only rationale given for favoring the possibility that "fast-swim singers" are uber-males is the assumption that there must be some way that listening whales are sorting singers based on their fitness, and that this is the only feature of fin whale singing that seems to vary across individuals in a way that the authors think could be used by listening whales to judge individuals. The fact that this possibility is conceivable provides no support for it.

Response: We agree with this reviewer that there are many imagined scenarios that might explain our observations of swimming and singing in fin whales. As the reviewer rightly states: The fact that a possibility "is conceivable provides no support for it." So how would a young fin whale singer possibly learn "that singing while rapidly swimming is a waste of time and energy?" We did not make or allude to the assumption that "there must be some way that listening whales are sorting singers based on their fitness, and that this is the only feature of fin whale singing that seems to vary across individuals in a way that the authors think could be used by listening whales to judge individuals." Rather, we are reporting on a unique set of observations describing fin whale swimming speeds and amount of singing. The data from these observations show that amount of singing is inversely related to swimming speed. Since metabolic and energetic cost is a function of swimming speed to the 3rd power in other marine mammal and at least one cetacean, we assume that for fin whales swimming fast is energetically costly and propose that fast swimming is an indication of stamina.

Given that all these observations happened in the ocean and half the time at night, the only way that another whale could detect if an animal is swimming fast is if that animal is acoustically active, with the probability of detection and range of detection being directly related to the sound's source level and propagation conditions. By scrutinizing details of each individual singer's speeds and amount of singing, we found a strong bimodal duty-cycle distribution for 1208 individual track segments, and that 69% of all singers swam at speeds greater than both the population average and median. These results certainly contradict the conclusion that swimming fast while singing is an abnormal behavior. How and whether or

not other whales are listening to and evaluating the swimming speeds of these singers is not known with certainty, but there are several studies showing that this is possible and we provide those references. Given the logistical difficulties of observing these animals, to our knowledge there are no observations of an interaction between a fin whale singer and a known female. Observations of a possible interactions between a fin whale singer and another male are anecdotal at best, and all we can contribute on that subject are a handful of observations that seemed to indicate acoustic interactions between two singers.

5. Reviewer-2, Item 5: The authors suggest that the capacity to sing while rapidly swimming might be a reliable indicator of fitness, but no measures of intra-individual variability of duty cycles produced during fast (or slow) swimming were presented in the report, so there is no way to assess whether any individual whale consistently sings at some maximum duty cycle while swimming fast. The presented data are consistent with the possibility that on some days singers will sing while fast swimming, but on other days they will not. The authors assume that whales that swim fast without singing would be singing while swimming if they were capable, but the authors present no evidence that the singers are not capable of doing so. If intra-individual variability of song production across days is comparable to inter-individual variability, then fast-swim singing would be useless as a fitness cue.

Response: We agree that our original interpretation of our observations on swimming speed and singing, which depended on duty-cycle, was erroneous. Our revised and simpler interpretation no longer assumes that high duty-cycle, a variable descriptor for amount of singing, is a behavioral feature by which listening animals assess male stamina or quality. We do not now assume that whales that swim fast without singing would be singing while swimming if they were capable. We do conclude, given the energetic cost of swimming, that whales that swim fast while singing, independent of how much they sing, could be advertising their stamina.

- a. We have added data on swimming speeds for individual track segments as shown in figure 3. This figure shows a strong bimodal duty-cycle distribution for 1208 individual track segments: a high duty-cycle mode (75-90% duty-cycle range, 34% of all track segments), with track segment speeds distributed between 1-8 km/h and a low duty-cycle mode (0-5% duty-cycle range, 21% of track segments), with track segment speeds distributed primarily between 1-10 km/h. Our first assumptions are that the singers we observed are males and that swimming speed is directly proportional to energetic cost. Both of these assumptions are supported by scientific evidence. From this we assume that

a male singer's swimming speed is a measure of his stamina. We further propose that listening whales can assess a male's swimming speed if he sings, but we do not assume that the amount of singing is a criterion by which listeners are making that assessment. This may or may not be the case (that is to say, amount of singing may turn out to be a feature indicative of stamina), but swimming speed alone is a reliable measure of a singer's stamina. Our hypothesis does by inference assume that there is a selective advantage to fin whales that can detect and track acoustically active conspecifics.

- b. Given the addition of song duration as a variable for amount of singing, we now present intra-individual variability of duty-cycle and song duration as a function of swimming speed.
- c. Results show that both duty-cycle and song duration decreased significantly as swimming speed increased.

6. Reviewer-2, Item 6: The idea that singing is energetically costly for baleen whales has been considered before in the literature, particularly in the case of singing humpback whales (e.g., Chu, 1988). Generally, there is no reason to believe that the energetic cost of singing is high. For instance, there is no evidence that dive times of singers are shorter than those of non-singers, which might be expected if singing were effortful. The fact that whales can sing almost continuously for periods of 24+ hours further suggests that singing does not require a large energy expenditure. Singing while swimming fast might be difficult for reasons unrelated to the amount of energy required. For example, the mechanics of sound production might be affected by rapid tail movements, or if a whale were swimming closer to the surface and blowing more frequently. The assumption that singing is energetically costly is not supported by any available data from baleen whales and is unnecessary to account for the observed behavioral correlation.

Response: We agree that “there is no reason to believe that the energetic cost of singing is high.” and have amended the text throughout the document accordingly. We went back into the very original data set and extracted 8023 song durations for 163 tracks (see figure 6). We included an additional analysis using average song duration as an independent variable. Song duration decreased with swimming speed as in the duty-cycle analysis. A strong seasonal peak was observed in both variables, as well as a weaker trend across years.

7. Reviewer-2, Item 7: The authors assume that other whales are monitoring and tracking singers over periods of hours or more. It is unclear how long a listener would need to monitor a singer to be able

to make a decision about the singer's capacity to sing while swimming fast. What is the probability that a listener would be in range of a singer long enough to make such a judgment? More generally, it is unclear that other whales are doing anything like this. The proposed scenario is highly speculative and assumes that listening fin whales possess several auditory and memory capacities that have never been shown in any species, including humans.

Response: It is true that we cannot specifically state what listening whales are doing in order to monitor a singer's speed. We did not attempt to model for the probability that a listener would be in range of a singer long enough to make such a judgment as there is not enough empirical data on fin whales to conduct such an exercise. Although there is only anecdotal evidence to support the conclusion that fin whales can respond to a singer, and there are no specific data regarding their auditory capacity and memory, we are very surprised, in fact a bit shocked, by the reviewer's last sentence, especially regarding the lack of evidence that humans can acoustically track other humans. Humpback whales and southern right whales can orient toward and approach playback sources to within a few meters (Mobley et al. 1988, Clark and Clark 1980). What about the classic work by Roger Payne demonstrating that owls can successfully catch rodents in complete darkness? What about the later work initiated by Konishi (1973) demonstrating the neurophysiology and neuroanatomy of the central nervous system by which owls accomplish this and other acoustically mediated critical life functions? What about the research on dolphins in captivity?

8. Reviewer-2, Item 8: If the authors' preferred interpretation of their findings were presented as such in the conclusions, and noted as being a post-hoc envisioning of how the observed phenomenon might relate to song function, then it would not be problematic. The current presentation of the observations instead is likely to give many readers the misimpression that the data provide evidence supportive of the authors' interpretation. For example, in the abstract, the authors state that, "The attributes of male acoustic advertisement displays are often related to a performer's age, breeding condition and motivation. We examined this relationship in fin whales, measuring a singer's swimming speed and amount of singing." This is not an accurate portrayal of the analysis that was performed, as it did not relate to age, breeding condition, motivation, or any other feature of an acoustic display that has been studied in any other acoustically displaying animal. The second sentence implies that swimming speed and singing duty cycle are known to be reproductively relevant attributes of songs, which is not the case.

Response: We agree with the reviewer's criticism of the over-interpretation in our last version of the manuscript, and we have attempted to avoid such speculation and over-interpretation.

Except for the first generic sentence in the abstract we have not pursued this idea in this paper, and we have removed the second sentence as quoted here: “We examined this relationship in fin whales, measuring a singer’s swimming speed and amount of singing.”

9. Reviewer-2, Item 9: Later in the abstract, the authors state that, “Given the high intensity of song notes and long durations of song bouts, we assume that singing is energetically costly. This suggests that the combination of swimming speed and singing rate is an endurance display by which females and/or other males can assess a singer’s physical fitness and potential reproductive quality.” Even if singing is energetically costly, which has yet to be established, this would not imply that fast-swim singing is an endurance display, because there is no evidence that fast-swim singing is difficult over the time spans that singer are doing it, or that it requires any more endurance than robust singing. This description presents a series of assumptions about a possible scenario with no empirical support as if it directly followed from the observations that the authors report.

Response: We have removed the assumption that singing is energetically costly. We do maintain that swimming is energetically costly. The only modality available to send a signal about their movement is through acoustics. Some individuals sing while swimming faster and some sing while moving slowly. The hypothesis that singing while swimming faster may indicate male quality is still valid, though remains unproven.

10. Reviewer-2, Item 10: I believe that the paper would be greatly improved by presenting the findings in a way that emphasizes the last point made in the abstract that, “Results have implications for interpreting fin whale singing behavior and the possible influences of anthropogenic sounds on fin whale mating strategies and breeding success.” The main significance of the observations lies in what they reveal about what fin whales are doing when they’re singing. Given the durations of the reported singing bouts, it seems likely that produced songs are effective regardless of a singer’s average swimming speed.

Response: Although we agree with the import of our results in terms of fin whale acoustic ecology, here we wanted to focus on the basic observation on swimming speed and amount of singing and not emphasize the potential implications of anthropogenic influences. In part, we did this because we have some evidence of whale swimming-singing behavior under different noise condition including anthropogenic.

11. Reviewer-2, Item 11: The example of non-robust song shown in Figure 2, if representative, suggests that non-robust singing is more temporally irregular, and Figure 3 seems to suggest that these

temporal irregularities might relate to changes in swimming speed. This finding seems similar to earlier reports that irregular structure in humpback whale songs is related to variability in dive times (Chu & Harcourt, 1986). It seems likely that the dive pattern of the singer shown in Figure 3 would be less regular than that of the singer shown in Figure 1. The authors cannot tell for sure when the singers they are tracking were surfacing, but they could still speak to the possibility that decreased dive times or less regular dive times might be correlated with changes in swimming speed, and the possibility that variations in dive duration might partly compensate for any increases in energy expenditure and/or account for intermittency in singing bouts.

Response: This is a very interesting suggestion! We decided not to speculate on this topic, but agree it would be a possible area of research, especially if one could track singers for entire bouts.

12. Reviewer-2, Item 12: Insufficient data is shared to support the article results. I assume this is because the data is from the SOSUS array. It would be preferable if the authors could at least provide audio files for the samples that they show in Figures 1-4.

Response: We agree with this suggestion. The audio files used to make figures 1 and 2 will be added to the dryad repository. We do not have access to the original acoustic data for what are now figures 4 and 5.

Literature Cited

- Agler, B. A., Schooley, R. L., Frohock, S. E., Katona, S. K., and Seipt, I. E. (1993). "Reproduction of photographically identified fin whales, *Balaenoptera physalus*, from the Gulf of Maine," *Journal of Mammalogy* **74**, 577-587.
- Braithwaite, J. E., Meeuwig, J. J., and Hipsey, M. R. (2015). "Optimal migration energetics of humpback whales and the implications of disturbance," *Conservation Physiology* **3**.
- Chu, K. C. (1988). "Dive times and ventilation patterns of singing humpback whales (*Megaptera novaeangliae*)," *Canadian Journal of Zoology* **66**, 1322-1327.
- Chu, K., and Harcourt, P. (1986). "Behavioral correlations with aberrant patterns in humpback whale *Megaptera novaeangliae* songs," *Behavioral Ecology and Sociobiology* **19**, 309-312.
- Clark, C. W., and Gagnon, G. C. (2004). "Low-frequency vocal behaviors of baleen whales in the North Atlantic: Insights from IUSS detections, locations and tracking from 1992 to 1996," *Journal of Underwater Acoustics* **52**.
- Clark, C. W., and Clark, J. M. (1980). "Sound playback experiments with southern right whales (*Eubalaena australis*)," *Science* **207**, 663-665.
- Croll, D. A., Clark, C. W., Acevedo, A., Tershy, B., Floress, S., Gedamke, J., and Urban, J. (2002). "Only male fin whales sing loud songs," *Nature* **417**, 809.
- Davis, R. W., Williams, T. M., and Kooyman, G. L. (1985). "Swimming metabolism of yearling and adult harbor seals *Phoca vitulina*," *Physiological zoology* **58**, 590-596.
- Gallagher, C. A., Stern, S. J., and Hines, E. (2018). "The metabolic cost of swimming and reproduction in harbor porpoises (*Phocoena phocoena*) as predicted by a bioenergetic model," *Marine Mammal Science* **34**, 879-900.
- Geoghegan, J. L., Pirotta, V., Harvey, E., Smith, A., Buchmann, J. P., Ostrowski, M., Eden, J. S., Harcourt, R., and Holmes, E. C. (2018). "Virological Sampling of Inaccessible Wildlife with Drones," *Viruses* **10**.
- Hind, A. T., and Gurney, W. S. C. (1997). "The metabolic cost of swimming in marine homeotherms," *The Journal of Experimental Biology* **200**, 531-542.
- Holt, M. M., Noren, D. P., Dunkin, R. C., and Williams, T. M. (2015). "Vocal performance affects metabolic rate in dolphins: implications for animals communicating in noisy environments," *The Journal of Experimental Biology* **218**, 1647-1654.

Ketten, D. R. (1997). "Structure and function in whale ears," *Bioacoustics* **8**, 103-135.

Konishi, M. (1973). "How the Owl Tracks Its Prey: Experiments with trained barn owls reveal how their acute sense of hearing enables them to catch prey in the dark," *American Scientist* **61**, 414-424.

Mobley, J. R., jr, Herman, L. M., and Frankel, A. S. (1988). "Responses of Wintering Humpback Whales *Megaptera novaeangliae* to Playback of Recordings of Winter and Summer Vocalizations and of Synthetic Sound," *Behavioral Ecology and Sociobiology* **23**, 211-224.

Appendix C

General Comments:

The authors have improved the paper by qualifying some of the claims being made about the implications of the observed correlations between singing and swim speed and by expanding the analyses of data. If listeners are able to judge differences in singers' speeds, and if singers are consistently singing at speeds during intervals sufficiently long to establish their relative fitness levels, then maybe other whales can/do use that information to assess singers. However, I think the authors might find it challenging to rank the singers they analyzed in terms of their probable fitness/size/age, especially if they had to do this in real-time by listening to the signal recorded at a single hydrophone. The data analyses are sound and with minor adjustments the paper would be suitable for publication.

Specific Remaining Issues:

Swimming speed (referenced throughout the paper). Fin whales swimming faster than ~7 km/hr are classified by the authors as "faster" and there is an implication that faster swimming is associated with greater stamina. At the same time, In 79 seems to indicate that fin whales are known to swim at speeds of 10-16 km/hr "for days." Does this mean that they can maintain these speeds continuously for 24+ hours? If so, then it is not clear how singers would be able to demonstrate their stamina by swimming at lower speeds for shorter intervals. In general, it is not clear how long a singer would need to sing/swim at a certain speed before other whales might be impressed. Obviously, there is no way to determine this from the data being analyzed, but it might be useful to provide some range of likely values. For instance, in Chu's discussion of song duration as a possible fitness cue in singing humpback whales, noted in the paper, duration is a proxy for dive time/breath holding ability, in which case 30+ minute songs without any surfacing would start being impressive. It's unclear from the current discussion in the paper whether any swim speed less than 20 km/hr would be fast enough to demonstrate superior stamina. Perhaps by estimating energy expenditure at different speeds, it might be possible to figure out when singers would start reaching their limits of endurance.

Perceiving sound speed. I think that the authors might want to point out in the manuscript (as they did in their response to my review) that the only way a fin whale might judge the speed, movements, size or any other feature of a distant fin whale is by using sound. That being said, judging differences in the speed of a 18-28 Hz source underwater by listening to it from one location is a non-trivial perceptual task. This is the ability I was referring to in the previous review when I suggested that neither humans nor other species have shown the ability to perceive the absolute speed (actually velocity) of a singing whale underwater. Animals can of course localize and track sound sources, but this is not the same thing as perceiving the absolute speed of a sound source. Imagine closing your eyes while a jogger runs by humming. Would you be able to state the runner's absolute speed from hearing them run by? Now imagine there are multiple humans humming while running in a race to demonstrate their stamina, can you hear which one is running the fastest? Possibly, but even then you probably would not be able to identify the speed the winner was running from hearing hums. Approaching (or avoiding) a sound source is a trivial task by comparison. Humans can barely tell the difference between a speech source at 70 m and one at 100 m even when nothing is

moving. Maybe fin whales can use their super-long wavelength sounds to judge each other's absolute speeds, but no other non-whales can do this that I know of and no reports demonstrating such an ability are cited in this paper for either non-whales or whales.

The paragraph starting at line 359 attempts to address this issue but oversimplifies the perceptual problem by implying that if listeners can track a singer, they will automatically know the singer's speed. The authors can calculate absolute speeds using tracks because they can calculate locations, exact durations, and distances using widely spaced receivers and can integrate across these measures over relatively long-time intervals. Perceiving movement is different from identifying a specific speed. Range-resolution for low-frequency sounds would be critical for estimating speed. It would be a simpler more defensible discussion if the authors just noted that whales' ability to judge differences in speed through audition are unknown but must be acute for songs to provide useful stamina cues of the sort being proposed.

Bimodal duty cycles. I found this new result more intriguing than the original speed-singing correlations because the distribution seems to be so bimodal. Why the big divide? It seems to suggest that 5-8 km/h might be a sweet spot for "fast sporadic singing" while 2-4 km/h is best for robust singing, and that these might correspond to a few typical singing modes. The finding seems similar to that reported by Henderson, Helble et al. (2018) for humpback whales in which the main singer tracks they found were of singers "milling"/diving or travelling in a specific direction.

Minor suggestions:

In 91, "peaked" should be "piqued"

In 300, this comparison is a bit unclear because humpback whales weren't necessarily swimming while singing, and the variation in song duration was used as to estimate dive times/breath-holding ability. Breath-holding could be a stamina cue. The link would be that maybe whales use sounds to show what they are physically capable of; this connection could be made clearer.

In 310, "high quality singer" sounds like the song is high quality – maybe "high quality mate?"
In 318, not clear what relatively close proximity means for two singing fin whales; 500 m?
In 386, this statement is a bit misleading because humans do not have the ability to accurately estimate the range to a low Hz source, or to judge the absolute speed of such sources.

Appendix D

Review of “Fin Whale Singing Decreases with Increased Swimming Speed”, Clark, Gagnon and Frankel,
Royal Society Open Science

General Comments received on 2/11/19, 10:42 AM

Reviewer comments to Author, received on 2/11/19, 10:42 AM

General Comments

Reviewer-1 (as converted from PDF file): The authors have improved the paper by qualifying some of the claims being made about the implications of the observed correlations between singing and swim speed and by expanding the analyses of data. If listeners are able to judge differences in singers' speeds, and if singers are consistently singing at speeds during intervals sufficiently long to establish their relative fitness levels, then maybe other whales can/do use that information to assess singers.

However, I think the authors might find it challenging to rank the singers they analyzed in terms of their probable fitness/size/age, especially if they had to do this in real-time by listening to the signal recorded at a single hydrophone. The data analyses are sound and with minor adjustments the paper would be suitable for publication.

Reviewer-2: In this revision, the authors have taken care to address the concerns of myself and the other original reviewer. As before, this work clearly demonstrates systematic differences in fin whale song production depending on swimming speed. The additions related to song duration strengthen the data set. The work is important and scientifically sound, and the analyses are appropriate.

The suggestion that listeners can assess male fitness based on swimming speed (with singing as the mechanism for determining speed) is interesting. While still speculative, the authors do a much better job of presenting their hypothesis as just that, and acknowledging their assumptions. Whether song production might also be an indicator of male quality remains to be seen, and the manuscript is simpler and more compelling without this additional assertion. I do think the discussion section could be shortened and made even simpler/more concise for clarity, with more emphasis on the direct meaning of the results rather than just the interesting potential implications.

Based on the authors' revisions, I can now recommend this manuscript for publication.

Authors' Responses to General Comments by Reviewer-1 and Reviewer-2

As with the earlier reviews, there were many valuable reviewer comments that challenged our earlier thinking and analysis. We took these seriously and have amended and revised the manuscript accordingly. We have added a supplemental table to help simplify the results section. We have reduced portions of the Discussion section and revised parts of the text so as to not over interpret our results. The text has now been shortened by almost 3 pages.

Specific Comments from Reviewer-1 with authors' responses (Note that Reviewer-1 used page and line numbers in the reviewed manuscript. The authors' responses use page and line numbers in the revised manuscript.)

Swimming speed (referenced throughout the paper). Fin whales swimming faster than ~7 km/hr are classified by the authors as “faster” and there is an implication that faster swimming is associated with greater stamina. At the same time, ln 79 seems to indicate that fin whales are known to swim at speeds of 10-16 km/hr “for days.” Does this mean that they can maintain these speeds continuously for 24+ hours? If so, then it is not clear how singers would be able to demonstrate their stamina by swimming at lower speeds for shorter intervals. In general, it is not clear how long a singer would need to

sing/swim at a certain speed before other whales might be impressed. Obviously, there is no way to determine this from the data being analyzed, but it might be useful to provide some range of likely values. For instance, in Chu's discussion of song duration as a possible fitness cue in singing humpback whales, noted in the paper, duration is a proxy for dive time/breath holding ability, in which case 30+ minute songs without any surfacing would start being impressive. It's unclear from the current discussion in the paper whether any swim speed less than 20 km/hr would be fast enough to demonstrate superior stamina. Perhaps by estimating energy expenditure at different speeds, it might be possible to figure out when singers would start reaching their limits of endurance.

Response: The fin whale referenced swimming at speeds of 10-16 km/hr "for most of four days." was not singing (Watkins, W. A. 1981), and there are no reports of tagged animals maintaining such speeds for 24+ hours. We feel that interpretation of speeds for fin whales from previous studies and singers as presented in this manuscript is a distraction from our results because the contexts were very different (a transiting, non-singer of unknown sex versus a singing male). This raises the challenging question as to how a non-singing male would demonstrate his stamina; however, in this revised manuscript we now include analysis and results comparing speeds for the same animal when singing and not singing, and report that overall there was no significant difference (methods lines 191-201, results lines 267-271). We decided not to speculate on what might constitute an impressive swimming speed for fin whale singers, but we agree that an obvious future avenue for exploring a stamina indicator would be via the energy-speed relationship that indicates that energetic cost is proportional to swimming speed to the third power (Hind, A. T., Gurney, W. S. C. 1997). We reduced discussion sections in which we had tried to speculate how listeners might possibly assess singer speeds, and we do not attempt to engage in discussions about what speed might be considered "impressive" or "fast enough" to a listener.

Perceiving sound speed. I think that the authors might want to point out in the manuscript (as they did in their response to my review) that the only way a fin whale might judge the speed, movements, size or any other feature of a distant fin whale is by using sound. That being said, judging differences in the speed of a 18-28 Hz source underwater by listening to it from one location is a non-trivial perceptual task. This is the ability I was referring to in the previous review when I suggested that neither humans nor other species have shown the ability to perceive the absolute speed (actually velocity) of a singing whale underwater. Animals can of course localize and track sound sources, but this is not the same thing as perceiving the absolute speed of a sound source. Imagine closing your eyes while a jogger runs by humming. Would you be able to state the runner's absolute speed from hearing them run by? Now imagine there are multiple humans humming while running in a race to demonstrate their stamina, can you hear which one is running the fastest? Possibly, but even then you probably would not be able to identify the speed the winner was running from hearing hums. Approaching (or avoiding) a sound source is a trivial task by comparison. Humans can barely tell the difference between a speech source at 70 m and one at 100 m even when nothing is moving. Maybe fin whales can use their super-long wavelength sounds to judge each other's absolute speeds, but no other non-whales can do this that I know of and no reports demonstrating such an ability are cited in this paper for either non-whales or whales.

The paragraph starting at line 359 attempts to address this issue but oversimplifies the perceptual problem by implying that if listeners can track a singer, they will automatically know the singer's speed. The authors can calculate absolute speeds using tracks because they can calculate locations, exact durations, and distances using widely spaced receivers and can integrate across these measures over relatively long-time intervals. Perceiving movement is different from identifying a specific speed. Range-resolution for low-frequency sounds would be critical for estimating speed. It would be a simpler more defensible discussion if the authors just noted that whales' ability to judge differences in speed through audition are unknown but must be acute for songs to provide useful stamina cues of the sort being proposed.

Response: We have revised text in the discussion (see lines 332-337) so as to read “Our results are consistent with but insufficient to assert our hypothesis that swimming speed is an indicator of a male’s physical stamina (i.e. a performance display). We affirm that singing is the mechanism by which listeners can directly assess swimming speed, while assuming that listening whales can somehow judge differences in singer swimming speeds and that the combination of swimming while singing provides the mechanism by which listeners assess a singer’s physical fitness.” We no longer delve into the topic of whether or not and how fin whale listeners might assess or compare singer swimming speeds, as that would become too speculative and well beyond the scope of this paper.

Bimodal duty cycles. I found this new result more intriguing than the original speed-singing correlations because the distribution seems to be so bimodal. Why the big divide? It seems to suggest that 5-8 km/h might be a sweet spot for “fast sporadic singing” while 2-4 km/h is best for robust singing, and that these might correspond to a few typical singing modes. The finding seems similar to that reported by Henderson, Helble et al. (2018) for humpback whales in which the main singer tracks they found were of singers “milling”/diving or travelling in a specific direction.

Response: We agree that this was an intriguing result. We appreciated the alert to the Henderson et al. 2018 paper and use that reference in our revision. We have slightly modified Figure 3 such that the cell count values have been converted to \log_{10} values, per the right-hand greyscale bar. We did this to better show the variability in the relationship between swimming speed and duty-cycle when considering all 1208 track segments.

Minor Comments from Reviewer-1 with authors’ responses (Note that Reviewer-1 used page and line numbers in the reviewed manuscript. The authors’ responses use page and line numbers in the revised manuscript.)

In 91, “peaked” should be “piqued”.

Response: In 95: Thank you! This misspelling has been corrected

In 300, this comparison is a bit unclear because humpback whales weren’t necessarily swimming while singing, and the variation in song duration was used to estimate dive times/breath-holding ability. Breath-holding could be a stamina cue. The link would be that maybe whales use sounds to show what they are physically capable of; this connection could be made clearer.

Response: This is a good point. The text has been revised (Ln 304-3112) so as to clarify this distinction. It now reads: “These results are consistent with the general notion that baleen males use features of their singing behavior as a mechanism by which to advertise their stamina and physical condition. For humpback whales, one hypothesis has been that song duration, which in some cases can be linked to breath-holding and directly related to a singer’s physical condition, could be an indication of male quality (Chu and Harcourt 1986). Likewise, significant difference in song unit frequency and amplitude parameters between individuals has also been suggested as a potential indicator of singer quality (Frankel 19943). For fin whales, a possible relationship between features of singing behavior and physical stamina raise a number of interesting questions as to the import of our empirical observations.”

In 310, “high quality singer” sounds like the song is high quality – maybe “high quality mate?”

Response: Ln 319: The text was amended to read “high-quality mate”.

Ln 318, not clear what relatively close proximity means for two singing fin whales; 500 m?

Response: Ln 332: The text was modified in order to clarify our assessment of the term relatively close proximity. It now reads: “relatively close acoustic proximity to each other (i.e. < 10 km).”

Ln 386, this statement is a bit misleading because humans do not have the ability to accurately estimate the range to a low Hz source, or to judge the absolute speed of such sources.

Response: The sentence referencing humans ("non-acoustic specialists") has been removed from this paragraph.

Specific Comments from Reviewer-2 with authors' responses

I do think the discussion section could be shortened and made even simpler/more concise for clarity, with more emphasis on the direct meaning of the results rather than just the interesting potential implications.

Response: We agreed with this comment and revised the manuscript so as to make the text simpler and more concise, which also reduced the length of the text by almost 3 pages.

Minor Comments from Reviewer-2 with authors' responses

P 8, L 166. Effect should be affect.

Response: Ln 165: Thank you. The word was corrected.

P 8, L 178-179. Why was a duty cycle >70% defined as “robust?” Was this an arbitrary distinction?

Response: Ln 177-178: Text was amended to read: “A duty-cycle $\geq 70\%$ was subjectively considered “robust singing” (figure 1), and a duty-cycle < 70% was subjectively considered “intermittent singing” (figure 2)”

P 9, L 198-200. Should this information be provided in the Results rather than the Methods section?

Response: We agree with this suggestion and moved the last two sentences in this paragraph that presented results into the results section (Ln 267-271). We also now include Figure_S-1, which shows the distribution of all inter-song-intervals ($n = 7877$) for 163 singers. These are the biexponential curves fit to the log frequency distribution used to calculate the bout ending criteria (BEC) value of 35 minutes. By this criterion, a singer sang multiple bouts if there was an inter-song-interval in his track ≥ 35 min.

P 9, L 208. Remove comma after “initially.”

Response: L 210: Thank you, this comma was deleted.

P 11, L 251. As I am reading this, I think it might be helpful to have a box somewhere that includes simple definitions of the various relevant variables that the reader could refer to. This could include, for example: track segment, bout, song track, etc.

Response: We seriously considered this suggestion, but decided not to include a side bar of definitions. We did make an effort to simplify and clarify terms, and to use them consistently throughout the text.

P 12, L 275-276. The first part of this sentence is unclear. What do the authors mean by “after the first 7.3 h of his sound bout compared to the following 8.7 h of the bout?”

Response: L 266-268: Both paragraphs presenting results for the two example speed versus duty-cycle tracks have been simplified. This previous section (P 12, L 275-276) has been rewritten to such that this previous sentence has been deleted.

P 12, L 279. Suggest change “at a” to “as his.”

Response: L 265-266: This text was amended to read “This whale was able to maintain some level of singing at a maximum speed of 19 km/h.”

P 13, Discussion section. This section is still fairly lengthy, and dedicated mainly to the potential interpretation/implications of the data. A lot of new background information is presented – could any of this be included in the Background section?

Response: Good suggestion. We have reduced and simplified the Discussion section by at least 2 pages so as to reduce speculation about how listeners might assess singer speeds.

P 13, L 297-298. The second sentence of the discussion doesn’t really seem to follow from the first. It seems that the authors are trying to make the point that the males with the highest physical fitness (and thus those able to swim 23 km/h while singing) are the ones singing while swimming faster than the average whale? This could be presented more clearly. It also might be helpful if the authors spent a bit more time at the beginning of the discussion summarizing the main results, what they mean, and why they are valuable in and of themselves, rather than jumping right in to talk about the potential implications of the results and future (interesting!) research questions.

Response: Good suggestions, and we have reorganized and rewritten most of the discussion accordingly.

P 14, L 321-325. This section regarding future research directions is a great addition here.

Response: Thank you.

P 15, L 338. Suggest change to “...and singing would provide the modality...”

Response: L 336: Text modified to read “singing would provide the mechanism”

P 17, L 387. Should read “...possess an equal...”

Response: This paragraph has been revised and the sentence containing this text has been deleted.